# FreeMo: Motion Generation with Structured Joint-Collision Energy

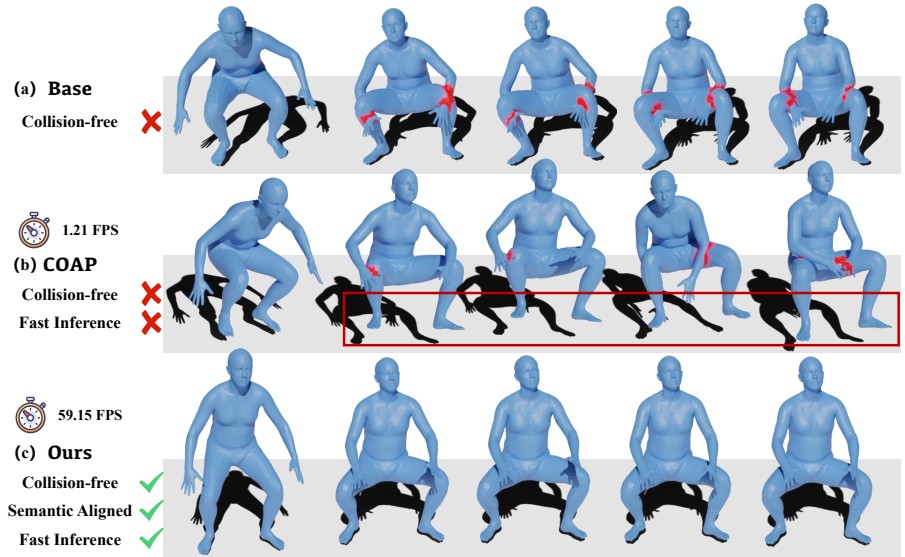

*The person sits and rests both wrists on their thighs.*

Figure 1: **Overview of challenges and motivation.** (a) The baseline model generates motions that exhibit severe self-collisions and lack physical plausibility. (b) Post-hoc correction using COAP reduces collisions but breaks semantic alignment and suffers from slow inference. (c) Our method, FreeMo, produces collision-free, semantically aligned motions with fast inference by integrating differentiable, trajectory-level self-collision constraints directly into the generation process.

## ABSTRACT

In this paper, we present FreeMo, a motion generation framework that produces physically plausible human motion by explicitly addressing self-collisions, where body parts intersect in unrealistic ways. Existing physics-aware generation models primarily handle external interactions, such as foot-ground contact, but are not capable of managing internal body dynamics. Although self-collisions can be corrected using post-hoc methods, these approaches are computationally expensive, difficult to scale, and compromise the differentiability and editability of the generative process. FreeMo integrates structured spatiotemporal constraints into the diffusion sampling process through a differentiable trajectory-level energy function that detects and penalizes persistent joint-level collisions. By directly optimizing joint positions in the latent space, FreeMo guides the generation away from physically implausible motions without compromising semantic alignment or motion naturalness. Experimental results show that FreeMo consistently reduces self-collisions while maintaining high-quality, controllable, and efficient motion synthesis.

## 1 INTRODUCTION

Text-driven human motion generation (T2M) aims to synthesize plausible human motions from natural language descriptions. Recent progress has been driven by deep generative models that learn mappings between text and motion. Variational frameworks (Petrovich et al., 2022), autoregressive Transformers (Zhang et al., 2023a; Guo et al., 2024a), and especially diffusion-based models (Zhang

et al., 2024; Tevet et al., 2022; Dai et al., 2024; Hong et al., 2025) have all advanced the generation quality in terms of semantic alignment, diversity, and temporal coherence.

However, despite improvements in visual fidelity, generated motions often suffer from physically implausible artifacts such as foot sliding, floating limbs, or ground penetration. Several recent works (Yuan et al., 2023; Han et al., 2025) have attempted to address these issues by incorporating physics-based corrections into the generation or post-processing pipeline. These approaches have shown promise in reducing interpenetration with the external environment, such as enforcing foot-ground contact or realistic dynamics. Yet, one important aspect remains underexplored: self-collision, where different parts of the body intersect in physically invalid ways (see Fig. 1a).

Existing efforts to handle self-collision are typically found in adjacent domains such as pose estimation or 3D reconstruction. Techniques like part-based implicit representations (Mihajlovic et al., 2022; 2025) or differentiable flow fields (Davydov et al., 2024) enable post-hoc correction of collisions, but are not designed to be integrated into the motion generation process. While it is technically possible to apply these post-hoc techniques to correct generated motions after sampling, doing so undermines the flexibility, interpretability, and editability of the generation process. Once motion is finalized and corrected in a non-differentiable manner, it becomes difficult to trace or manipulate the relationship between the text prompt and the resulting motion. Moreover, as these methods operate frame-by-frame without modeling temporal dynamics, applying them across sequences is computationally expensive and prone to unnatural transitions (see Fig. 1b). This is further compounded by the need for frequent prompt tuning during generation, making such approaches impractical for scalable use.

An alternative to post-hoc correction is to guide the generative process directly. In the image domain, classifier-guided diffusion methods (Nichol & Dhariwal, 2021; Kim et al., 2022a;b) have shown that external signals can help steer sampling toward semantically aligned outputs. However, applying similar strategies to motion generation is challenging, as classifier feedback tends to be coarse and unstable in continuous spaces like motion trajectories.

A more principled approach is to use differentiable energy functions that provide structured, continuous feedback during sampling. Prior works (Yu et al., 2023; Hong, 2024; Zhang et al., 2025; Ron et al., 2025) have successfully integrated such energy terms into diffusion-based models to enforce semantic structure, compositional control, and physical constraints, especially in the context of human-environment interaction. These methods demonstrate that energy-based guidance can enhance generation quality while maintaining compatibility with gradient-based optimization. However, existing energy functions have primarily focused on high-level semantics, making these ideas not directly applicable to the problem of self-collision.

Self-collisions within the human body present a more localized and temporally dynamic challenge. They often occur between specific joint pairs and persist over time, requiring the model to reason not only about spatial proximity but also about motion trajectories. A simple frame-by-frame proximity penalty fails to capture this complexity. For example, during walking, a natural arm swing may cause the hand to pass near the torso or leg. Penalizing this proximity uniformly across frames would suppress natural motion and introduce stiffness. What matters is whether the proximity is prolonged and physically implausible, not whether it occurs momentarily in isolation. Addressing self-collisions thus requires constraints that consider not only spatial relationships between body parts but also their evolution over time.

To this end, we introduce **FreeMo**, a unified self-collision-aware motion generation framework that improves diffusion-based synthesis by incorporating a spatiotemporal understanding of joint-level interactions. At the core of FreeMo is a differentiable, trajectory-based energy module that penalizes persistent collisions between joints over time. Unlike conventional approaches that rely on mesh-based (Klosowski et al., 1998), volumetric (Mihajlovic et al., 2025), or SDF-based (Mihajlovic et al., 2022) proximity detection, which are difficult to integrate into generative models due to their non-differentiability and computational cost, FreeMo operates directly on joint trajectories. This design ensures compatibility with gradient-based sampling while maintaining efficiency and interpretability.

FreeMo is more than a loss function. It is a generation framework that unifies physical constraints with data-driven motion synthesis. By modeling collision tendencies across entire motion sequences, it dynamically adjusts the sampling trajectory in response to self-collision risk. At the

same time, it preserves alignment with the input text, allowing for the generation of natural, expressive, and semantically faithful motions (see Fig. 1c). Because FreeMo integrates collision awareness directly into the generative process, it avoids the limitations of post-hoc correction, and supports scalable, editable, and controllable motion generation across a wide range of prompts. *Video results are provided in the supplementary materials.*

We summarize our contributions as follows:

- We propose the first self-collision-aware motion generation framework that integrates structured spatiotemporal constraints into the diffusion sampling process.
- We design the Structured Joint-Collision Energy Function, a differentiable, trajectory-level energy module that detects and penalizes persistent joint-level collisions, improving physical plausibility without degrading semantic alignment or motion quality.
- Experiments demonstrate that FreeMo significantly reduces self-collisions while preserving motion naturalness, controllability, and sampling efficiency.

## 2 RELATED WORK

**Text-to-motion Generation.** Recent advances in text-to-motion generation are dominated by diffusion models. MotionDiffuse (Zhang et al., 2024) and MDM (Tevet et al., 2022) improve quality via CLIP conditioning and classifier-free guidance. Latent models such as MLD (Chen et al., 2023), MotionLCM (Dai et al., 2024), and SALAD (Hong et al., 2025) enhance efficiency and controllability, while retrieval-augmented and GPT-style methods (e.g., ReMoDiffuse (Zhang et al., 2023b), T2M-GPT (Zhang et al., 2023a), MotionGPT (Jiang et al., 2023)) improve diversity and coherence. Yet, most models neglect physical constraints, resulting in artifacts like foot sliding and self-collision. We address this by introducing a differentiable, trajectory-aware energy that enables self-collision-free motion generation without retraining.

**Physics-aware Motion Generation.** Recent methods incorporate physical constraints to improve motion realism. PhysDiff (Yuan et al., 2023) uses an in-loop physics projection to enforce contact and balance, while ReinDiff (Han et al., 2025) applies reinforcement learning for physically plausible training. BioMoDiffuse (Kang et al., 2025) introduces biomechanical priors based on joint torques and energy, and UniPhys (Wu et al., 2025) combines planning and control in a unified diffusion framework. Diffuse-CLoC (Huang et al., 2025) adds look-ahead control for future-aware generation. Unlike them, we target intra-body interactions by embedding joint-level collision constraints into diffusion optimization, enabling efficient self-collision avoidance without simulation or retraining.

**Post-hoc Self-collision Removal.** Traditional solutions (Li & Barbič, 2018; Nesme et al., 2009; Sifakis et al., 2007; Karras, 2012; Tzionas et al., 2016) to self-collision are slow and incompatible with generation. Learning-based approaches such as COAP (Mihajlovic et al., 2022), CLOAF (Davydov et al., 2024), and VolumetricSMPL (Mihajlovic et al., 2025) improve efficiency but remain post-hoc and introduce latency or optimization instability. We instead embed joint-level collision awareness into the diffusion process, enabling efficient, temporally coherent, and semantically faithful motion synthesis.

## 3 METHOD

### 3.1 PROBLEM FORMULATION

Given a natural language prompt $\mathcal{T}$, our goal is to generate a human motion sequence $M = \{\hat{\mathbf{x}}_{t,j}\}$, where $\hat{\mathbf{x}}_{t,j} \in \mathbb{R}^3$ denotes the global position of joint $j$ at frame $t$. To obtain SMPL parameters, we apply a non-differentiable inverse kinematics (IK) solver (Voleti et al., 2022) to each pose. The resulting SMPL poses are used to deform a human mesh sequence $V = \{V_t\}_{t=1}^L$ via the SMPL model (Loper et al., 2023), where $V_t \in \mathbb{R}^{V \times 3}$ denotes the 3D mesh at frame $t$. We focus on generating motions that are both semantically aligned with the input text and free from self-collisions. Formally, the generative model $G$ maps a text prompt to motion:

$$M = G(\mathcal{T}), \quad V_t = \text{SMPL}(IK(M_t)). \tag{1}$$

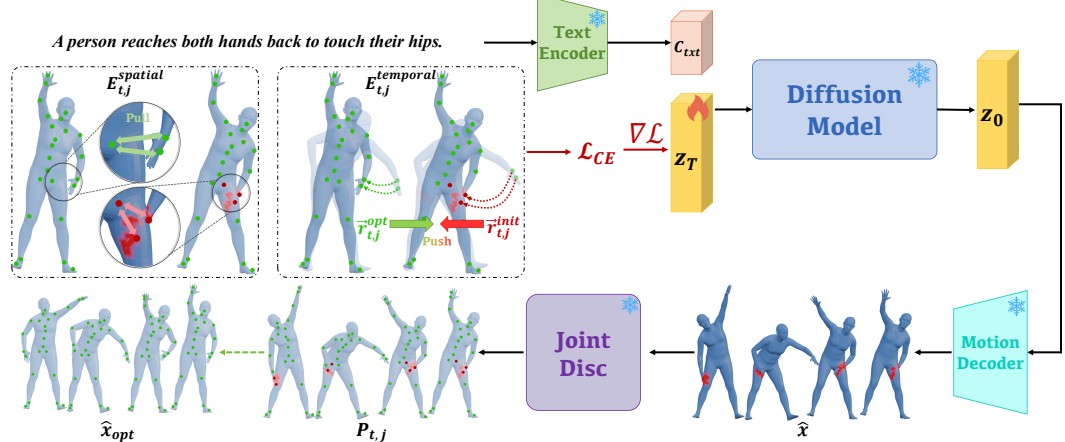

Figure 2: **Overview of our FreeMo framework.** Given a text prompt, a pretrained diffusion model produces a latent representation $z_T$, which is decoded into a motion sequence. A joint-level discriminator estimates per-joint collision probabilities $P_{t,j}$, and our Structured Joint-Collision Energy Function evaluates the severity of each collision via a weighted sum of spatial and temporal terms: $E_{t,j} = \alpha_s E_{t,j}^{\text{spatial}} + \alpha_t E_{t,j}^{\text{temporal}}$. The combined signal $P_{t,j} E_{t,j}$ is used to guide inference-time optimization of $z_T$, producing refined motions that are collision-free and semantically faithful to the input prompt.

Self-collisions are defined as the presence of non-adjacent triangle pairs intersecting in 3D space. Let $\mathcal{F} = \{(i_1, i_2, i_3) \mid (i_1, i_2, i_3) \in \mathcal{V}^3\}$ be the fixed face topology of the SMPL mesh, where each tuple $(i_1, i_2, i_3)$ represents a triangle defined by vertex indices from the vertex set $\mathcal{V}$. For frame $t$, a triangle is denoted:

$$f_o^t = \triangle \left( V_t^{(i_1)}, V_t^{(i_2)}, V_t^{(i_3)} \right), \quad \text{where } (i_1, i_2, i_3) \in \mathcal{F}. \tag{2}$$

We require that for all $t \in \{1, \dots, L\}$, no two non-adjacent triangles intersect:

$$\forall t, \quad \nexists (f_i^t, f_j^t) \in \mathcal{F} \times \mathcal{F}, \, i \neq j, \quad f_i^t \cap f_j^t \neq \emptyset. \tag{3}$$

This constraint ensures that the generated human mesh is free from physically implausible self-collisions throughout the motion sequence.

## 3.2 SELF-COLLISION OPTIMIZATION

Directly optimizing joint positions to remove self-collisions often leads to unnatural artifacts, such as stiff or semantically misaligned motions. This is primarily due to joint-level changes are highly sensitive and may disrupt the overall motion structure. For instance, MoManifold (Aytekin et al., 2025) shows that naive pose-level regularization disrupts temporal dynamics and collapses motions into near-static trajectories. Similarly, Pose-NDF (Tiwari et al., 2022) argues that the pose space lacks an inherent human plausibility prior, necessitating projection onto a learned manifold to maintain realism. These observations indicate that optimization in pose space is inherently unstable and semantically fragile, frequently producing unrealistic or misaligned motions.

A closely related idea is Diffusion Noise Optimization (DNO) (Karunratanakul et al., 2024), which adjusts the input noise $x_T$ in the original diffusion space to satisfy constraints without retraining the model. Formally:

$$x_T^* = \arg\min_{x_T} \mathcal{L}(\text{ODESolver}(d(\cdot), x_T, \mathcal{T})), \tag{4}$$

where $d$ is a pretrained text-to-motion diffusion model, ODESolver denotes the denoise sampler, and $\mathcal{L}$ is a differentiable constraints.

While DNO performs optimization on the original noise input $x_T$ of a diffusion model, our method targets the latent noise $z_T$ used in latent-based diffusion models such as MLD (Chen et al., 2023),

MotionLCM (Dai et al., 2024), and SALAD (Hong et al., 2025). This shift to the latent space facilitates optimization in a more compact and semantically structured form, enabling more efficient and stable refinement. However, leveraging this advantage requires more than simply applying standard optimization techniques. To effectively eliminate self-collisions without degrading motion quality, the optimization process must be guided by an objective that is both structured and differentiable, capturing the spatial configuration and temporal dynamics of joint interactions.

## 3.3 STRUCTURED JOINT-COLLISION ENERGY FUNCTION

To effectively and stably guide the optimization process, we propose a Structured Joint-Collision Energy Function, a differentiable joint-level objective tailored for latent-based diffusion models.

Let $z_T$ be the latent noise at the end of the diffusion process. A generated motion sequence $M = \{\hat{\mathbf{x}}_{t,j}\}$ is obtained by feeding $z_T$ through the deterministic denoising steps followed by a motion decoder. Our energy function is designed to guide the optimization of $z_T$ such that the decoded motion becomes self-collision free while preserving semantic fidelity. Following COAP, we define joint collision as the case where a joint lies inside the occupancy region of another body part. We train a joint-level discriminator $\mathbf{P}$, which takes a single-frame pose vector as input and predicts the collision probability for each joint. At inference time, the pretrained discriminator is used to compute per-joint collision probabilities $\mathbf{P}_{t,j} = \mathbf{P}(\hat{\mathbf{x}}_{t,j}) \in [0,1]$ from each frame's pose vector. If $\mathbf{P}_{t,j} > \tau$, the joint is considered at risk of collision and will be penalized. The overall loss is:

$$\mathcal{L}_{\text{CE}} = \frac{1}{N} \sum_{t=\Delta+1}^{T-\Delta} \sum_{j=1}^{J} \mathbf{I}[\mathbf{P}_{t,j} > \tau] \cdot \mathcal{L}_{\text{BCE}}(\mathbf{P}_{t,j}, 0) \cdot E_{t,j}, \tag{5}$$

where $\mathcal{L}_{\text{BCE}}$ is the binary cross-entropy loss with ground truth label 0, indicating that joints should not be colliding. The energy term $E_{t,j}$ measures the severity of potential collision from both spatial and temporal perspectives:

$$E_{t,j} = \alpha_s \cdot E_{t,j}^{\text{spatial}} + \alpha_t \cdot E_{t,j}^{\text{temporal}}. \tag{6}$$

The spatial term penalizes proximity to other joints:

$$E_{t,j}^{\text{spatial}} = \exp\left(-\frac{\min_{k \neq j} \|\hat{\mathbf{x}}_{t,j} - \hat{\mathbf{x}}_{t,k}\|^2}{\sigma_0^2}\right), \tag{7}$$

and the temporal term discourages abrupt changes from the original joint trajectories:

$$\vec{r}_{t,j} = \hat{\mathbf{x}}_{t+\Delta,j} - \hat{\mathbf{x}}_{t-\Delta,j},$$
$$E_{t,j}^{\text{temporal}} = 1 - \frac{\vec{r}_{t,j}^{\text{opt}} \cdot \vec{r}_{t,j}^{\text{init}}}{\left\|\vec{r}_{t,j}^{\text{opt}}\right\|_2 \left\|\vec{r}_{t,j}^{\text{init}}\right\|_2}, \tag{8}$$

where $\Delta$ indicates the window size of a joint trajectory, $\vec{r}^{\text{opt}}$ is computed from the current motion, and $\vec{r}^{\text{init}}$ is from the motion decoded using the initial $z_T$. This term ensures that the refined motion remains consistent with the initial semantics. We incorporate our Joint-Collision Energy $\mathcal{L}_{\text{CE}}$ into the DNO framework by treating it as the guidance loss during latent optimization:

$$z_T^* = \arg\min_{z_T} \mathcal{L}_{\text{CE}}(\text{Dec}(\text{ODESolver}(d(\cdot), z_T, \mathcal{T}))). \tag{9}$$

Through iterative updates of $z_T$, the final generated motion becomes physically plausible while maintaining its original semantics.

## 4 EXPERIMENTS

### 4.1 DATASETS AND EVALUATION METRICS

**HardPoseText Benchmark.** We construct a benchmark named *HardPoseText*, designed to evaluate motion generation models under scenarios prone to self-collisions. The benchmark comprises challenging textual prompts that often result in entangled or compact human poses. Initially, 20 representative prompts were selected from the HumanML3D dataset (Guo et al., 2022). Using GPT (Brown et al., 2020), we expanded these into 200 complex motion descriptions, exemplified

by cases such as "kneeling down while hugging one's knees." For each prompt, five motion sequences were synthesized using MDM (Tevet et al., 2022), and the average self-collision rate was calculated. The 50 prompts exhibiting the highest mean penetration are selected as the final *HardPoseText* benchmark, and each text prompt is assigned a random sampling length between 40 and 200. This curated dataset serves as a rigorous testbed for assessing model performance in high-risk self-collision contexts. Additional details, including the GPT construction template, diversity analysis, and cross-model difficulty ranking, are provided in Appendix C.

**HumanML3D Subset.** To assess generalization to normal prompts, we use the first 1000 samples from the HumanML3D (Guo et al., 2022) test split. This test set comprises diverse and standard text descriptions of human actions, enabling us to validate both generalization and motion preservation.

**Evaluation Metrics.** To measure self-collisions, we propose metrics based on the occupancy representation proposed in COAP (Mihajlovic et al., 2022). Specifically, we first identify pairs of body parts whose axis-aligned bounding boxes overlap, and uniformly sample 3D points within the overlapping volumes. Each body part $p$ has a learned occupancy function $\mathcal{O}_p(\cdot) \in [0, 1]$, predicting whether a 3D point lies inside the spatial region of part $p$. For each sampled point $\mathbf{q}_{t,k}$ at frame $t$, we compute its occupancy value as the maximum occupancy across all body parts excluding its owner:

$$o_{t,k} = \max_{p \neq \text{owner}(\mathbf{q}_{t,k})} \mathcal{O}_p(\mathbf{q}_{t,k}) \tag{10}$$

The *Collision Score (Col.Score)* is defined as the average occupancy value over all sampled points and frames:

$$\text{Col.Score} = \frac{1}{T} \sum_{t=1}^{T} \sum_{k=1}^{K} o_{t,k} \tag{11}$$

To compute the *Collision Rate (Col.Rate)*, we define a binary indicator that specifies whether each sampled point is in collision:

$$c_{t,k} = \begin{cases} 1, & \text{if } o_{t,k} > 0.01 \\ 0, & \text{otherwise} \end{cases} \tag{12}$$

The Collision Rate is then defined as the percentage of frames containing at least one colliding point:

$$\text{Col.Rate} = \frac{1}{T} \sum_{t=1}^{T} \mathbf{1} \left[ \sum_{k=1}^{K} c_{t,k} > 0 \right] \tag{13}$$

These metrics provide continuous and discrete measures of self-collision severity by aggregating occupancy statistics over spatial samples in potential collision regions. To evaluate motion quality and diversity of the generated motion, we follow the standard protocol of MDM (Tevet et al., 2022). Specifically, we report *R-Precision (Top-3)*, and *Multimodal Distance (MM-Dist)* to assess the semantic alignment between generated motions and their corresponding input prompts. To quantify motion realism, we compute the *Fréchet Inception Distance (FID)* in the motion feature space. To assess the generative diversity, *Diversity* is computed as the overall variance across generated samples, while *MultiModality (MModality)* measures the average variance conditioned on a single text prompt. Following prior work (Yi et al., 2022), motion smoothness is quantified using *Jitter*, defined as the mean change in joint acceleration over time, measured in units of $10^2$ m/s$^3$. To evaluate computational efficiency, we report *milliseconds per frame (ms/frame)*, calculated as the average time required to generate a single frame during inference, using the maximum batch size supported by the GPU. All experiments are conducted on a single NVIDIA L40S GPU. All reported results are averaged over 20 independent runs.

## 4.2 SELF-COLLISION EVALUATION

**Quantitative Evaluations.** We evaluate base models and our approach on the proposed HardPoseText benchmark in terms of self-collision. As shown in Table 1, our method achieves a dramatic reduction in self-collision metrics across all base models. Notably, the collision rate on MotionLCM decreases from 31.35% to 1.44%, while on MLD it is reduced from 23.09% to 0.49%. The collision score also shows a significant drop, confirming the effectiveness of our joint-level optimization. In addition to improving physical plausibility, our method slightly enhances semantic alignment and motion smoothness. For instance, R-Precision (top 3) increases from 0.231 to 0.262 on MLD, and

Table 1: **Comparisons on HardPoseText benchmark.** All reported results are averaged over 20 independent runs. ↓ indicates lower is better, ↑ indicates higher is better. **Bold** indicates the best performance within each group. Our method significantly reduces self-collision while preserving motion quality.

| Method | Col. Rate ↓ | Col. Score ↓ | R-Precision (top 3) ↑ | MM-Dist ↓ | Jitter ↓ | ms/frame ↓ |
|---|---|---|---|---|---|---|
| MLD | 23.09 | 50.58 | 0.231 | 5.138 | 0.39 | **1.71** |
| MLD + Ours | **0.49** | **0.70** | **0.262** | **4.997** | **0.22** | 39.75 |
| MotionLCM | 31.35 | 84.41 | 0.318 | **4.205** | 3.96 | **0.94** |
| MotionLCM + COAP | 21.33 | 17.59 | 0.245 | 5.020 | 5.02 | 826.45 |
| MotionLCM + Ours | **1.44** | **1.44** | **0.331** | 4.488 | **0.85** | 16.95 |
| SALAD | 20.94 | 45.37 | 0.291 | **4.581** | 0.38 | **1.23** |
| SALAD + Ours | **5.97** | **9.79** | **0.312** | 5.154 | **0.30** | 30.36 |

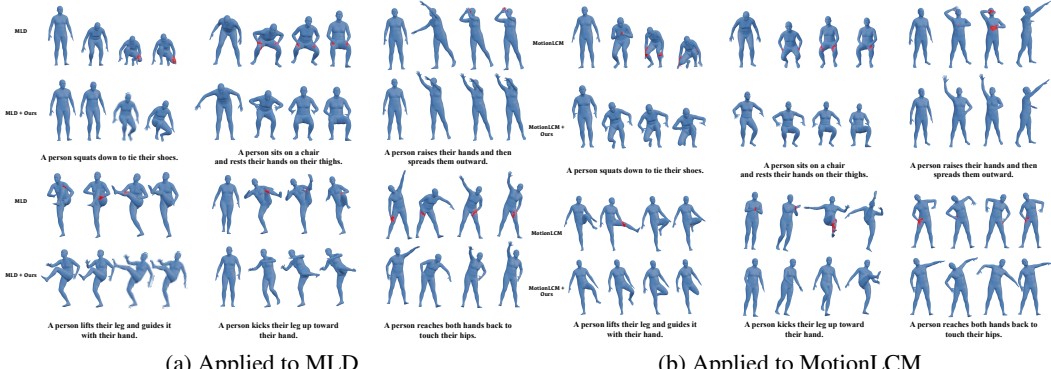

(a) Applied to MLD          (b) Applied to MotionLCM

Figure 3: **Qualitative results generated by (a) MLD and (b) MotionLCM, w/o and w/ our approach.** Our method effectively eliminates self-collisions (highlighted in red) while preserving natural motion and intended semantics across diverse prompts. Zoom in for better views.

jitter decreases from 3.96 to 0.85 on MotionLCM, without any explicit jitter loss. We attribute this to the temporal consistency term in our energy function, which discourages abrupt trajectory changes and encourages smooth motion transitions. On SALAD, the reduction in self-collision is comparatively less dramatic (from 20.94% to 5.97%), which we believe is due to the skeleton-aware latent space used in SALAD. This compact representation limits the granularity of per-joint correction during latent-space optimization, thereby reducing the optimization flexibility. Nevertheless, our method consistently improves both collision metrics and motion diversity across all baselines, demonstrating its general applicability across different architectures.

Importantly, the computational overhead remains modest, especially in light of the significant performance gains. As reported in Table 1, our method remains orders of magnitude faster than post-processing approaches such as COAP, which are memory-intensive and inefficient for sequential data. To further contextualize these results, we evaluated COAP on MotionLCM for the HardPose-Text benchmark. COAP fails to resolve collisions effectively and introduces substantial motion jitter, while our method achieves lower collision rates (1.44 vs. 21.33), reduced jitter (0.85 vs. 5.02), and over 40× faster processing. It also delivers higher-quality motion, reflected in R-Precision (0.331 vs. 0.245) and MM-Dist (4.488 vs. 5.020). Although CLOAF claims temporal optimization, its lack of publicly available code prevents direct comparison. Overall, *FreeMo* provides a unified solution that simultaneously improves motion quality, physical plausibility, and computational efficiency.

**Qualitative Results.** The qualitative results are presented in Figure 3a, Figure 3b, and Figure 4, which show the outputs of three representative baseline models: MLD, MotionLCM, and SALAD, each compared with and without our method. Across all examples, FreeMo consistently removes self-collisions while preserving the intended motion semantics. Notably, in prompts such as "A person sits on a chair and rests their hands on their thighs" and "A person reaches both hands back to touch their hips", our method maintains natural hand-body contact after optimization, demonstrating its ability to distinguish meaningful contact from undesired collisions. These results highlight the robustness and general applicability of FreeMo across different generative backbones, confirming

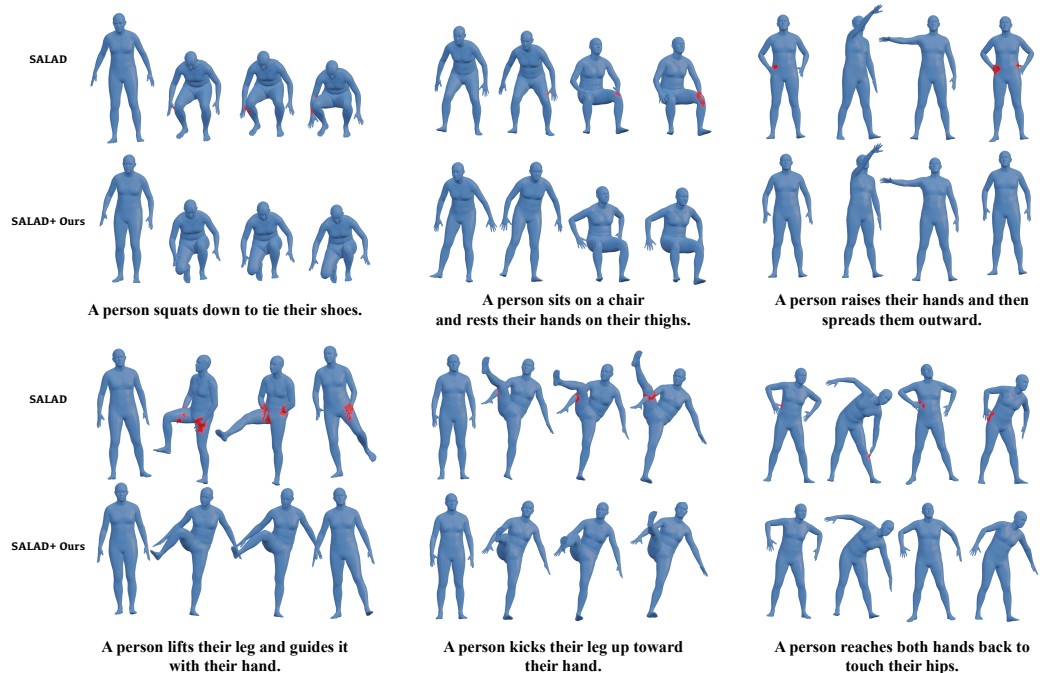

Figure 4: **Qualitative results generated by SALAD, w/o and w/ our approach.** Similar to Figure 3, our method removes self-collisions while preserving motion quality.

Table 2: **Generalization and motion preservation evaluation on HumanML3D subset.** All results are averaged over 20 independent runs. ↓ indicates lower is better, ↑ indicates higher is better, and → indicates closer to real data is better. **Bold** highlights the best performance within each group. This table evaluates both motion quality and self-collision metrics, showing that our method reduces collisions even in simpler motion scenarios while consistently preserving motion quality.

| Method | Col. Rate ↓ | Col. Score ↓ | R-Precision (top 3) ↑ | MM-Dist ↓ | FID ↓ | Diversity → | MModality ↑ |
|---|---|---|---|---|---|---|---|
| Real | - | - | 0.800 | 2.926 | - | 9.486 | - |
| MLD | 10.93 | 16.17 | **0.824** | **2.790** | 0.158 | 9.669 | 1.644 |
| MLD + Ours | **2.11** | **1.71** | 0.814 | 2.833 | **0.154** | **9.513** | **1.782** |
| MotionLCM | 7.92 | 7.60 | **0.829** | **2.760** | **0.153** | 9.566 | 1.769 |
| MotionLCM + Ours | **1.95** | **2.87** | 0.802 | 2.944 | 0.310 | 9.289 | **1.995** |
| SALAD | 6.95 | 8.21 | **0.858** | **2.619** | **0.122** | 9.602 | 1.798 |
| SALAD + Ours | **1.13** | **1.29** | 0.830 | 2.986 | 1.664 | **9.420** | **2.085** |

its effectiveness in diverse motion contexts and input conditions. *Video results are provided in the supplementary materials.*

## 4.3 GENERALIZATION AND MOTION PRESERVATION EVALUATION

Table 2 reports results on the HumanML3D subset, evaluating both motion quality and self-collision metrics across different base models. While most metrics reflect motion quality only, we additionally include collision rate and collision score for completeness. Compared to the HardPoseText benchmark, the improvements in self-collision metrics here are less pronounced. This is expected, as HumanML3D primarily contains simpler actions such as standing, walking, or waving (e.g., "A person waves their hand"), which naturally result in fewer self-collisions. As a result, the baseline collision rates are already relatively low compared to Table 1. Nevertheless, *FreeMo* consistently reduces both collision rate and score across all models, effectively narrowing the remaining gap toward zero. More importantly, this reduction in collisions does not come at the cost of motion quality. In fact, *FreeMo* often preserves or enhances key aspects of motion semantics and variation. For instance, on MLD, *FreeMo* maintains high R-Precision (0.814 vs. 0.824) and increases

Table 3: **Ablation study on different loss terms. Bold** indicates the best performance. Our loss design effectively resolves the self-collision issue, maintaining high-quality and natural motion.

| Base | Components | Col.Rate ↓ | Col.Score ↓ | R-Prec. ↑ | MM-Dist ↓ | Jitter ↓ |
|------|-----------|------------|-------------|-----------|-----------|----------|
| MotionLCM | – | 31.35 | 84.41 | 0.318 | **4.205** | 3.96 |
| MotionLCM | Global D | 12.71 | 23.15 | 0.254 | 5.210 | 1.03 |
| MotionLCM | Joint D | 4.70 | 7.61 | 0.293 | 4.237 | 0.79 |
| MotionLCM | Joint D + T | 7.80 | 6.57 | **0.343** | 4.554 | **0.51** |
| MotionLCM | Joint D + S + T | **1.44** | **1.44** | 0.331 | 4.488 | 0.85 |

D = Discriminator, S = Spatial Energy, T = Temporal Energy

MModality (1.644 to 1.782). On MotionLCM, it achieves the highest MModality (1.995) and a strong R-Precision of 0.802. Even for SALAD, with its compact latent space, *FreeMo* improves MModality (1.798 to 2.085) while slightly reducing Diversity. These results confirm that *FreeMo* generalizes well across architectures, reduces self-collisions even in easier motion contexts, and preserves or improves motion quality.

## 4.4 ABLATION STUDY

We conduct an ablation study in Table 3 to evaluate the impact of different loss components.

**Effect of collision discriminator loss.** Starting from the baseline without any discriminator (Row 1), adding a global discriminator (Row 2) significantly reduces the collision rate and score. However, it degrades motion quality, as indicated by a higher MM-Dist and lower R-Precision. This variant uses a global binary classification loss $\mathcal{L}_{\text{global}} = \mathcal{L}_{\text{BCE}}(P_t, 0)$, where $P_t$ is the predicted collision probability for the entire pose at frame $t$.

Replacing the global discriminator with a joint-level variant (Row 3) achieves a better trade-off. It further lowers the collision rate (4.70%) and jitter (0.79), while maintaining competitive motion quality (MM-Dist 4.237). This variant predicts per-joint collision probabilities $P_{t,j}$ and uses a joint-wise loss $\mathcal{L}_{\text{joint}} = \sum_{j=1}^{J} \mathcal{L}_{\text{BCE}}(P_{t,j}, 0)$. Given its stronger balance between physical plausibility and motion quality, the joint-level discriminator is adopted as the default in our framework.

**Effect of temporal energy term.** Adding the temporal energy term (Row 3 vs. Row 4) leads to a notable reduction in jitter (0.51, the lowest among all) and improves R-Precision (0.343). However, collision metrics worsen slightly, suggesting that temporal-only penalties suppress motion magnitude and smooth out motion artifacts but are insufficient to fully resolve self-collisions.

**Effect of spatial energy term.** Introducing the spatial energy term (Row 4 vs. Row 5) achieves the lowest collision rate (1.44%) and collision score (1.44), while maintaining competitive R-Precision (0.331) and MM-Dist (4.488). Although jitter increases slightly compared to the temporal-only setting (0.85 vs. 0.51), the combination of spatial and temporal energy terms proves complementary. Together, they enforce physical plausibility without degrading semantic alignment, avoiding overly rigid motion or naive joint separation.

## 4.5 SENSITIVITY ANALYSIS.

We further examine how the key hyperparameters, including the discriminator threshold $\tau$ and the spatial and temporal energy weights $(\alpha_s, \alpha_t)$, influence the behavior of our method. The purpose of this analysis is to verify that the optimization remains stable when these values are changed, and to understand whether the overall motion quality is sensitive to their magnitudes.

**Sensitivity on HardPoseText.** We begin with the HardPoseText benchmark, where self-collisions occur frequently. As shown in Table 4, changing $\tau$ produces the expected monotonic trend: a smaller threshold responds to potential collisions earlier, while a larger threshold delays intervention. Adjusting $(\alpha_s, \alpha_t)$ shifts the balance between spatial separation and temporal consistency, but the motion quality and semantic alignment remain stable across settings.

**Sensitivity on HumanML3D.** To check whether this behavior holds in simpler motion contexts, we repeat the same analysis on a HumanML3D subset. The results in Table 5 show similarly smooth

| $\tau$ | $\alpha_s$ | $\alpha_t$ | Col. Rate ↓ | Col. Score ↓ | R-Precision (top 3) ↑ | MM-Dist ↓ | Jitter↓ |
|---|---|---|---|---|---|---|---|
| 0.1 | 1.0 | 1.0 | 0.77 | 1.18 | 0.306 | 4.505 | 0.85 |
| 0.2 | 1.0 | 1.0 | 1.44 | 1.44 | 0.331 | 4.488 | 0.85 |
| 0.3 | 1.0 | 1.0 | 2.29 | 3.03 | 0.337 | 4.454 | 0.84 |
| 0.2 | 1.0 | 0.5 | 1.57 | 2.10 | 0.356 | 4.409 | 0.84 |
| 0.2 | 0.5 | 1.0 | 1.61 | 2.99 | 0.337 | 4.369 | 0.83 |

Table 4: Sensitivity analysis of $\tau$, $\alpha_s$, and $\alpha_t$ on MotionLCM evaluated on HardPoseText.

and predictable variations. Since HumanML3D contains fewer poses with intrinsic self-collisions, absolute collision values are smaller, but the effect of each hyperparameter remains consistent.

| $\tau$ | $\alpha_s$ | $\alpha_t$ | Col. Rate ↓ | Col. Score ↓ | R-Precision (top 3) ↑ | MM-Dist ↓ | FID ↓ | Diversity → | MModality ↑ |
|---|---|---|---|---|---|---|---|---|---|
| 0.1 | 1.0 | 1.0 | 1.23 | 1.46 | 0.803 | 3.021 | 0.478 | 9.419 | 2.308 |
| 0.2 | 1.0 | 1.0 | 1.95 | 2.87 | 0.802 | 2.944 | 0.310 | 9.289 | 1.995 |
| 0.3 | 1.0 | 1.0 | 1.96 | 3.06 | 0.832 | 2.858 | 0.280 | 9.699 | 2.102 |
| 0.2 | 1.0 | 0.5 | 1.87 | 2.37 | 0.822 | 2.911 | 0.320 | 9.573 | 2.224 |
| 0.2 | 0.5 | 1.0 | 1.97 | 3.12 | 0.826 | 2.897 | 0.317 | 9.589 | 2.226 |

Table 5: Sensitivity analysis of $\tau$, $\alpha_s$, and $\alpha_t$ on MotionLCM evaluated on HumanML3D subset.

## 5 CONCLUSION, LIMITATION, AND FUTURE WORK

We present FreeMo, a lightweight optimization framework for generating self-collision-free human motion from text. By integrating a structured, differentiable joint-collision energy into the diffusion noise optimization process, FreeMo enables collision-aware inference without modifying the base model, significantly reducing self-collisions while preserving motion quality and efficiency.

While FreeMo operates in joint level, its applicability to surface-level interactions is limited by current generators that lack mesh outputs. As mesh-based generators become feasible, we plan to extend FreeMo to support mesh-level collision handling for richer physical interactions.

## 6 REPRODUCIBILITY STATEMENT

To ensure reproducibility, we provide both our code and text prompts used in the HardPoseText benchmark in supplementary materials. For detailed implementation settings, please refer to Appendix E.

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

## A  THE USE OF LARGE LANGUAGE MODELS (LLMs)

In this work, LLMs were used solely as a language refinement tool to improve grammar, clarity, and readability of the manuscript, as well as to assist in the construction of prompts for HardPoseText Benchmark (see in Appendix C). All research ideas, methodologies, and experimental results were conceived, designed, and conducted by the authors without the involvement of the LLMs. No part of the scientific content, including problem formulation, analysis, or conclusions, was generated by the LLMs.

## B  OVERVIEW OF SUPPLEMENTARY MATERIALS

We provide additional benchmark construction details and analysis, qualitative results, implementation details, and environment information. All videos and source code are organized in the `supp/` directory.

- `supp/code` — inference optimization scripts, discriminator training code.
- `supp/HardPoseText.txt` — text prompts used in the HardPoseText benchmark.
- `supp/qualitative_results.mp4` — qualitative comparisons.

## C  HARDPOSETEXT BENCHMARK

### C.1  BENCHMARK CONSTRUCTION

To ensure that GPT-generated candidates remain consistent with the semantic space of HumanML3D, we use a constrained prompt template that enforces realistic human motions, simple syntax, and familiar everyday or athletic actions. The full template is shown below:

```
Here are several example text prompts from the
HumanML3D dataset, which describe realistic, everyday
human actions in simple natural language:
[representative prompts from HumanML3D]
Using the same writing style and naturalness, generate
new human motion descriptions that are:
1.  realistic and physically plausible motions that
humans perform in everyday or athletic contexts,
2.  likely to involve close body-part proximity or
self-contact (for example arms touching the torso,
legs crossing, curled poses, touching feet, reaching
behind the body),
3.  diverse across categories such as sitting,
kneeling, curling, stretching, twisting, crossing
limbs or reaching backward,
4.  expressed in one simple English sentence,
5.  not fantastical, not stylized, and not
anatomically impossible.
Generate N such motion descriptions in the same style
as HumanML3D. Output only the list of sentences.
```

This template restricts GPT to the same linguistic register and conceptual categories as HumanML3D. It does not introduce new semantic classes and serves only to broaden textual coverage toward realistic motions that naturally involve self-contact, which are underrepresented in the original dataset. For full transparency, we provide the full list of HardPoseText prompts and their corresponding motion lengths in the file `supp/HardPoseText.txt`, formatted as:

```
98 The person squats and rests their head on their
knees with arms wrapped around.
176 The person holds a tree pose with palms together
in front of the chest.
91 The person lifts one leg backward and leans
forward, forming a T-shape.
```

Each line contains the sampling length (number of frames) followed by the associated text prompt.

## C.2 PROMPT DIVERSITY ANALYSIS

To quantify prompt diversity, we use the semantic diversity metric based on Sentence-BERT (Reimers & Gurevych, 2019) embedding similarity, as introduced in the linguistic-diversity benchmark (Guo et al., 2024b).

| Dataset | Semantic Diversity ↑ |
| --- | --- |
| HumanML3D Test Set | 0.627 |
| HardPoseText | 0.513 |

Table 6: Prompt Diversity of HumanML3D and HardPoseText.

The results indicate that HardPoseText maintains a comparable level of dispersion across prompts, suggesting that the expanded dataset does not collapse into narrow lexical patterns and retains diversity consistent with established benchmarks.

## C.3 PROMPT BIAS ANALYSIS

To evaluate whether HardPoseText introduces any model-specific bias, we rank all prompts by collision rate using three representative motion generators (MLD, MotionLCM, SALAD). We then compute Spearman rank correlations between each pair of rankings:

| Model Pair | Spearman $\rho$ |
| --- | --- |
| MLD vs MotionLCM | 0.8068 |
| MLD vs SALAD | 0.3700 |
| MotionLCM vs SALAD | 0.4268 |

Table 7: Spearman rank correlations of HardPoseText prompt difficulty across different base models.

The correlations are consistently positive, and the strong agreement between MLD and MotionLCM suggests that prompts which induce collisions in one model tend to do so in others. The lower, yet positive, correlations with SALAD are expected given its compact latent structure. Overall, the results indicate that HardPoseText captures a model-agnostic notion of difficulty rather than reflecting biases toward any specific architecture.

**Discussion.** Although alignment scores on HardPoseText are lower than those on standard benchmarks, this is an inherent property of the dataset. The prompts intentionally describe compact or entangled poses where physically constrained motions are more challenging to generate. The reduced alignment therefore reflects the difficulty of the task rather than shortcomings in textual diversity or realism.

## D QUALITATIVE RESULTS

`supp/qualitative_results.mp4` shows qualitative comparisons between baseline models and our proposed method on a selection of challenging prompts from the HardPoseText benchmark, demonstrating the effectiveness of our approach in generating collision-free and natural-looking human motions, particularly in complex poses where self-collisions are more likely to occur. The video sequentially presents motion generation results from various baseline models (e.g., MLD,

MotionLCM, SALAD) alongside our method (denoted as "Ours") for the same text prompts. Each segment clearly labels the base model and our result for easy comparison. All motions are rendered at 15 FPS using SMPL meshes for visual consistency. The lower frame rate helps mitigate the risk of transient collision frames being missed during playback.

It is important to emphasize that the text prompts in the HardPoseText benchmark are highly complex and semantically nuanced. Even state-of-the-art motion generation models often struggle to generate motions that precisely reflect the detailed descriptions in the text. As a result, certain generated motions may not fully align with the intended semantics, and self-collisions may occur even when the overall pose appears plausible. For example, consider the prompt: "The person squats and rests both elbows on their knees while touching the ground". Both MLD and our method wrongly place the hands on the knees. However, our method effectively avoids collision between the hands and knees, a failure case that can commonly occur in baseline models. Similarly, for the prompt: "The person curls up into a tight ball on the floor", MLD generates a pose where the person is merely putting their hands and knees on the ground, as does our method. In this case, our approach avoids potential self-collisions between the elbows and knees that otherwise arise in MLD. When the base model demonstrates a reasonable understanding of the text descriptions but fails to address self-collisions, our generated motions do not exhibit such artifacts. For instance, in the prompt "The person bends over and rests their palms on the floor beside their feet", MLD produces a motion where the hands intersect each other. In "The person twists rapidly in place with arms extended", MLD generates foot-to-foot collisions. In "A person kicks their leg up toward their hand", SALAD results in a collision between the thigh and torso. And in "The person performs a handstand and holds it steadily" SALAD produces an arm-to-thigh penetration. These examples illustrate that even when the overall semantic of the motion is preserved, baseline models can produce physically implausible or visually unnatural motions due to self-collisions.

In several cases, our method not only avoids such collisions but also generates motions that better reflect the full intent of the text prompt. For the prompt "The person sits with their forehead on the floor and arms crossed beneath", MotionLCM generates a simple squatting motion. In contrast, our method produces a pose where the forehead is in contact with the ground and the arms are properly crossed underneath, closely matching the textual description. Similarly, for "The person balances on one knee and one hand, stretching the opposite limbs outward" and "The person touches the floor with one hand while extending the opposite leg backward" our generated motions exhibit a stronger alignment with the described actions compared to those from SALAD. These qualitative results demonstrate that our approach is effective in reducing self-collisions across a variety of challenging poses without sacrificing semantic faithfulness.

# E  IMPLEMENTATION DETAILS

## E.1  JOINT-LEVEL DISCRIMINATOR

We train a 6-layer MLP to classify whether each joint is in a self-colliding state. The input is a 263-dimensional pose vector extracted from each frame, and the output is a 44-dimensional tensor representing binary logits for 22 joints. The network consists of fully connected layers with hidden dimensions [1024, 512, 256, 128, 64, 32], with ReLU activation, BatchNorm, and a dropout rate of 0.1 after each layer. The final output is reshaped to $(22, 2)$ to represent per-joint binary logits.

To construct training data, we use a dataset denoted as `PosePeneSet`, generated by sampling 1000 motion clips from the HardPoseText benchmark using MDM (Tevet et al., 2022), with each prompt sampled 20 times. For each motion, we apply the COAP (Mihajlovic et al., 2022) method to detect collisions on every frame, obtaining per-joint collision scores. A joint is considered to be in penetration if its score exceeds 0.01. These binary labels are used as ground truth. The data is split into training, validation, and test sets in a 60%/20%/20% ratio. The model is trained using the Adam optimizer with a learning rate of 0.001, weight decay of 0.0001, and a cosine learning rate schedule with 5 warm-up epochs. Training runs for 100 epochs with a batch size of 64. The best-performing checkpoint on the validation set is selected for downstream use during inference-time optimization. On the validation set, the model achieves an accuracy of 97.8% with a precision of 81.2% and an F1 score of 73.5%.

### E.2  GLOBAL DISCRIMINATOR

For the global discriminator used in our ablation study, we adopt the same architecture as the joint-level model. The output is a 2-dimensional vector representing a binary classification of whether the entire pose frame contains any self-collision. To construct labels, we reuse the COAP-based annotations from `PosePeneSet`. A frame is labeled as "penetrated" if at least one joint is in a collision state. On the validation set, the model reaches 90.9% accuracy with a precision of 52.1% and an F1 score of 49.4%.

### E.3  COAP-BASED POST-PROCESSING

Given joint positions $\{\hat{\mathbf{x}}_{t,j}\}$ from the generated motion, we first apply an inverse kinematics (IK) solver (Voleti et al., 2022) to obtain the corresponding SMPL pose parameters $\theta \in \mathbb{R}^{T \times 72}$. We then refine these parameters by optimizing pose residuals $\Delta\theta$. At each step, the updated pose is $\theta_{\text{new}} = \theta + \Delta\theta$, and the mesh vertices $V_t = \{\mathbf{v}_{t,i} \in \mathbb{R}^3\}_{i=1}^{N_v}$ are computed via SMPL. The refined vertices are converted back to joint positions via the SMPL joint regressor, completing the correction process. We follow COAP (Mihajlovic et al., 2022) and uniformly sample a set of 3D query points within potentially colliding regions. For each sampled point $\mathbf{q}_{t,k}$, its occupancy is evaluated as:

$$o_{t,k} = \max_{p \neq \text{owner}(\mathbf{q}_{t,k})} \mathcal{O}_p(\mathbf{q}_{t,k}), \tag{14}$$

where $\mathcal{O}_p : \mathbb{R}^3 \to [0,1]$ is the learned occupancy field for part $p$. The self-collision loss is defined as:

$$\mathcal{L}_{\text{sc}} = \frac{1}{TK} \sum_{t=1}^{T} \sum_{k=1}^{K} o_{t,k}, \tag{15}$$

where $K$ is the number of sampled points per frame. We apply a regularization term on the pose residual:

$$\mathcal{L}_{\text{reg}} = \lambda_{\text{reg}} \|\Delta\theta\|_2^2, \tag{16}$$

and additionally adopt the same jitter loss as in the main paper (Eq. 16), defined as:

$$\mathcal{L}_{\text{jitter}} = \frac{1}{(T-2)J} \sum_{t=2}^{T-1} \sum_{j=1}^{J} \|\hat{\mathbf{x}}_{t+1,j} - 2\hat{\mathbf{x}}_{t,j} + \hat{\mathbf{x}}_{t-1,j}\|_2 \,, \tag{17}$$

The total loss is defined as:

$$\mathcal{L}_{\text{total}} = \lambda_{\text{sc}}\mathcal{L}_{\text{sc}} + \lambda_{\text{reg}}\mathcal{L}_{\text{reg}} + \lambda_{\text{jitter}}\mathcal{L}_{\text{jitter}}, \tag{18}$$

where we set $\lambda_{\text{sc}} = 1.0$, $\lambda_{\text{reg}} = 0.01$, and $\lambda_{\text{jitter}} = 0.01$. Optimization is performed using the Adam optimizer with $\beta_1{=}0.9$, $\beta_2{=}0.999$, a learning rate of 0.001, batch size 4 (i.e., 4 consecutive frames per sub-sequence), and a maximum of 400 steps. Early stopping is applied if $\mathcal{L}_{\text{sc}} < 10^{-5}$.

### E.4  ENVIRONMENT DETAILS

All experiments were conducted on a Linux server running Ubuntu 22.04 with kernel version 5.15. The system is equipped with one NVIDIA L40S GPU with 48GB of memory. All experiments utilized CUDA 12.1 and Python 3.10.

