# OpenReview forum: "FreeMo: Motion Generation with Structured Joint-Collision Energy"
_ICLR.cc/2026/Conference — Submitted to ICLR 2026_

### Official Review · Reviewer_zvWv · 2025-10-29

**Soundness:** 2
**Presentation:** 2
**Contribution:** 1
**Rating:** 4
**Confidence:** 5

**Summary:**

This paper introduces FreeMo, a framework designed to generate physically plausible human motion from text descriptions by explicitly preventing self-collisions. FreeMo integrates a novel, differentiable energy function directly into the diffusion sampling process, which detects and penalizes persistent, unrealistic intersections between body parts at the joint level.

**Strengths:**

Effective and Integrated Collision Avoidance.

High Efficiency and Quality Preservation.

**Weaknesses:**

- The main concern with this work is the limited contribution. The joint collision optimization has been widely explored by the community of geometry and animation, such as mesh recovery. The authors simply transfer this idea to motion generation and do not clarify the difference from the previous work. Besides, if the proposed method works and is novel, it can be used in other tasks, not only motion generation. Please try mesh recovery and music-driven motion generation.

- The work might lack real application. The output is an SMPL sequence, but not a real character in the industry. The collision optimization is useless in real applications.

- The method requires a latent optimization, which looks similar to that in MotionLCM and omnicontrl. What is the difference?

- The latent optimization looks time-consuming in inference.

**Questions:**

see weakness

---

> ### Author Response · Authors · 2025-11-23
>
> **Weaknesses**
>
> > W1. This work is a simple transfer from mesh recovery.
>
> We respectfully disagree with the claim that our method simply transfers ideas from mesh recovery to motion generation. This appears to stem from a misunderstanding of the core differences between the two problem settings and the technical requirements of modern generative models.
>
> Mesh-based collision handling methods in geometry and animation are inherently post-processing techniques. They operate after a motion sequence is fully generated, often relying on mesh-level proximity queries, signed distance fields, or iterative surface relaxation. These approaches are non-differentiable, computationally expensive, and frame-based. As a result, they cannot be integrated into diffusion sampling or provide gradients for latent-space guidance, which are essential for modern motion generation frameworks.
>
> In contrast, FreeMo addresses self-collision during the generative process itself. We propose a structured, trajectory-level joint-collision energy function that is differentiable with respect to the latent space. This enables gradient-based control during inference, allowing the generated motions to remain semantically faithful while reducing persistent collisions in a temporally coherent way. The approach is fundamentally different from surface-level correction, both in formulation and in how it interacts with the generative model.
>
> To our knowledge, no prior work from the geometry modeling literature has demonstrated a feasible way to reduce self-collisions in text-to-motion generation. If transferable ideas exist, we would sincerely welcome concrete suggestions. However, based on available methods and our empirical findings, such a transfer is far from straightforward and has not been achieved in existing work.
>
> FreeMo fills this gap by offering an efficient and differentiable framework that directly addresses self-collision within the generative pipeline. It is a principled solution to a long-standing challenge that remains under-addressed in generative motion research.
>
> > W2. The work might lack real application. The output is an SMPL sequence, but not a real character in the industry. The collision optimization is useless in real applications.
>
> We respectfully disagree with the claim that our work lacks practical application. This concern may stem from a geometry-oriented perspective, but it is important to distinguish between mesh reconstruction and motion generation. **The goal of motion generation is to eliminate the need for labor-intensive motion capture, which is of substantial value across industries including animation, gaming, virtual production, robotics, and AR/VR.** These applications often rely on joint-level motion representations, which are efficient, expressive, and easy to integrate into downstream systems. Our method is agnostic to character format. Although we use SMPL for benchmarking due to its research adoption, FreeMo operates entirely at the joint level and does not depend on any particular mesh. It can be directly adapted to other character systems as long as joint trajectories are available.
>
> We are unsure what "real application" the reviewer has in mind, but joint-based motion generation is already widely deployed in production settings. Our method focuses on **generating** physically plausible motion that avoids self-collisions during inference, which is a fundamentally different task from mesh recovery.
>
> We believe this comment underestimates the practical value and impact of motion generation as a field. Prioritizing mesh-centric views risks overlooking the growing number of applications where joint-level control, physical plausibility, and generative flexibility are critical.
>
> > W3. The method requires a latent optimization, which looks similar to that in MotionLCM and omnicontrl. What is the difference?
>
> We respectfully clarify that while FreeMo also uses latent optimization, its purpose and formulation are fundamentally different from MotionLCM and OmniControl.
>
> These prior methods apply latent tuning to adjust **controllability**, focusing on steering specific joints to target positions at certain times, but they do not address physical correctness. They lack any modeling of joint interactions, do not estimate penetration risk, and are not designed to enforce self-collision avoidance.
>
> In contrast, FreeMo is the first to address **physical feasibility** directly in latent space. Its optimization focuses not on changing what motion is generated, but on ensuring that the same motion is performed in a physically plausible, self-collision-free manner. This represents a distinct goal and mechanism from prior latent refinement techniques.

---

> ### Author Response · Authors · 2025-11-23
>
> > W4. The latent optimization looks time-consuming in inference.
>
> We agree that enforcing physical constraints during inference, particularly for reducing self-collisions, requires additional computation. This is a reasonable trade-off when improving physical feasibility. However, we respectfully disagree with the characterization of our method as "time-consuming".
>
> FreeMo performs optimization entirely in latent space, which is both efficient and low-dimensional. As shown in Table 1, it achieves a per-frame cost of 16.95 milliseconds on MotionLCM, which is suitable for real-time or interactive use. More importantly, to our knowledge, no existing method in either motion generation or mesh recovery offers comparable self-collision reduction at similar or lower inference cost. For example, COAP, a representative post-processing baseline, takes 826.45 (50x slower) milliseconds per frame and still fails to effectively eliminate collisions.
>
> In comparison, FreeMo strikes a favorable balance between efficiency and physical correctness. The additional computation is modest and well justified by consistent improvements in collision metrics, motion quality, and semantic alignment.
>
> **To Reviewer zvWv: We believe some concerns may stem from a geometry-oriented perspective, which differs from the objectives of motion generation. Our work focuses on generating physically plausible motion during inference, which remains an open problem in generative modeling, rather than on post-hoc mesh correction. To our knowledge, no existing method from geometry or motion generation achieves real-time self-collision-aware generation. We have clarified this distinction in our rebuttal and respectfully request a re-evaluation based on the context and goals of motion generation research.**

---

### Official Review · Reviewer_3FC3 · 2025-10-30

**Soundness:** 2
**Presentation:** 3
**Contribution:** 2
**Rating:** 6
**Confidence:** 2

**Summary:**

This paper introduces FreeMo, a framework for text-driven motion generation that addresses self-collision by integrating a Structured Joint-Collision Energy Function into the diffusion process. The method penalizes persistent joint collisions while preserving semantic alignment and motion quality. Experiments on the HardPoseText benchmark and HumanML3D subset demonstrate significant reductions in self-collision across multiple baselines, with competitive motion quality and efficiency.

**Strengths:**

- Tackles the underexplored issue of self-collision in motion generation.
- Embeds collision constraints directly into the generative process, avoiding inefficiencies of post-hoc corrections.
- Demonstrates effectiveness on diverse benchmarks with well-defined metrics.
- Well-written and organized with detailed experiments and analysis.

**Weaknesses:**

- Generated motions occasionally fail to fully align with text prompts, especially for complex descriptions.
- Focuses only on joint-level collisions, neglecting surface-level interactions.
- HardPoseText relies on GPT-expanded prompts, raising concerns about diversity and realism.
- Overemphasis on collision metrics; broader physical plausibility is underexplored.

**Questions:**

Please see Weaknesses for details.

---

> ### Author Response · Authors · 2025-11-23
>
> **Weaknesses**
> > W1. Generated motions occasionally fail to fully align with text prompts, especially for complex descriptions.
>
> We acknowledge that alignment issues may occur in complex prompts, but our analysis shows that these primarily originate from the base generative models rather than our method. On HardPoseText, all base models exhibit lower R-Precision and higher MM-Dist compared to HumanML3D, confirming their limited ability to handle challenging prompts.
>
> FreeMo does not introduce additional misalignment. Instead, it preserves the semantics of the original motion through temporal consistency and discriminator-guided latent updates. As shown in Table 2, semantic metrics remain stable or slightly improve after applying FreeMo. This indicates that our method retains the base model’s interpretation while effectively removing self-collisions.
>
> > W2. Focuses only on joint-level collisions, neglecting surface-level interactions.
>
> While our optimization operates in joint space for efficiency and differentiability, all evaluations of self-collision are conducted at the mesh level. Specifically, we follow the COAP protocol by reconstructing SMPL meshes from the optimized joint sequences and applying a volumetric occupancy-based metric to detect mesh interpenetration. Both Collision Rate and Collision Score are computed using mesh surface interactions, not joint proximity.
>
> Moreover, our joint-level discriminator is trained using mesh-based collision labels derived from COAP on the PosePeneSet dataset. This supervision ensures that joint-level predictions align with true mesh penetrations. As shown in our experiments (Table 1 & Table 2), FreeMo consistently reduces mesh-level collisions across multiple base models. This demonstrates that optimizing in joint space yields practical improvements in surface-level physical plausibility.
>
> > W3. HardPoseText relies on GPT-expanded prompts, raising concerns about diversity and realism.
>
> We respectfully clarify that HardPoseText was constructed with careful control over diversity and realism. The GPT-generated prompts were constrained by a strict template that preserves the semantic scope and linguistic simplicity of HumanML3D. This ensures the prompts remain realistic, grounded in everyday or athletic motions, and free from fantastical or anatomically implausible content. The inclusion of self-contact scenarios broadens coverage within the same motion domain rather than introducing new categories.
>
> GPT is used solely as a lexical and stylistic augmentation tool, not a semantic generator. The generated prompts are intentionally similar in tone and complexity to HumanML3D, as enforced by the template shown below.
>
> ```
> Here are several example text prompts from the HumanML3D dataset, which describe realistic, everyday human actions in simple natural language:
> [representative prompts selected]
> Using the same writing style and naturalness, generate new human motion descriptions that are:
>
> 1. realistic, physically plausible motions that humans perform in everyday or athletic contexts
> 2. likely to involve close body-part proximity or self-contact (e.g., arms touching torso, legs crossing, curled poses, touching feet, reaching behind the body)
> 3. diverse across categories, such as sitting, kneeling, curling, stretching, twisting, crossing limbs, reaching backward, etc.
> 4. described using one simple English sentence
> 5. not fantastical, not overly stylized, and not anatomically impossible
>
> Generate N such motion descriptions in the same style as HumanML3D.
> Output only the list of sentences.
> ```
>
> To quantify the diversity of HardPoseText prompts, for any two prompts $p_i$ and $p_j$, with embeddings $e_i$ and $e_j$, the pairwise diversity is defined as:
> $$
> div_{i, j} = 1 - cos_{sim}(e_i, e_j).
> $$
> The overall dataset diversity is obtained by averaging $div_{i,j}$ over all prompt pairs.
>
> | Dataset | Prompt Diversity↑ |
> | :-----: | :--------: |
> | HumanML3D Test Set| 0.627 |
> | HardPoseText | 0.513 |
>
> The results show comparable lexical dispersion across prompts, indicating that the expanded dataset maintains diversity levels similar to established benchmarks, rather than collapsing into narrow or repetitive textual patterns. While R-Precision scores on HardPoseText are lower than those on standard benchmarks, this is expected. The dataset is intentionally designed to be more challenging, with prompts that elicit compact or entangled poses prone to self-collisions. These lower alignment scores reflect the increased difficulty of generating physically plausible motion under constrained conditions, not deficiencies in prompt quality or diversity.

---

> ### Author Response · Authors · 2025-11-23
>
> > W4. Overemphasis on collision metrics; broader physical plausibility is underexplored.
>
> We would like to emphasize that self-collisions are a core aspect of physical plausibility that remains insufficiently addressed in generative motion models. These errors break anatomical realism and frequently occur even in high-quality outputs from current systems.
>
> While our focus is on self-collision, the method is evaluated comprehensively, including semantic fidelity (R-Precision), motion quality (MM-Dist), and temporal smoothness (Jitter). The results show that enforcing collision-free motion does not degrade overall quality, and in many cases improves it.
>
> More importantly, this work lays a foundation for broader physical reasoning. Joint-level, differentiable optimization enables real-time, semantics-aware corrections that traditional post-processing or physics-heavy solutions cannot offer. By solving a neglected but fundamental issue in motion generation, FreeMo opens a practical path toward richer, physically grounded generative models.

---

> > ### Comment · Reviewer_3FC3 · 2025-11-26
> > **RE: Rebuttal**
> >
> > Thank you for the response. While I am not an expert in this field, I think the authors have addressed most of the concerns I could think of. I will maintain my positive rating.

---

> > > ### Author Response · Authors · 2025-11-26
> > >
> > > We sincerely appreciate your follow-up comment and your continued positive assessment of our manuscript. Should any further questions or points requiring clarification arise, please do not hesitate to let us know. We would be pleased to provide any additional information that may be helpful.

---

### Official Review · Reviewer_ufHe · 2025-10-31

**Soundness:** 3
**Presentation:** 2
**Contribution:** 2
**Rating:** 4
**Confidence:** 4

**Summary:**

The paper addresses the problem of self-collisions in text-to-motion generation, where a character’s limbs or body parts interpenetrate unnaturally. Current text-driven motion diffusion models excel at semantic alignment but often ignore physical constraints, leading to artifacts like limbs passing through each other. FreeMo is proposed as the first motion generation framework explicitly designed to avoid self-collisions during generation.

The authors introduce FreeMo, a plug-in optimization framework that integrates Structured Joint-Collision Energy into the diffusion sampling process. Rather than retraining models, FreeMo performs inference-time latent optimization (inspired by Diffusion Noise Optimization) on a pre-trained motion diffuser’s latent code. A joint-level collision discriminator is trained to predict collision probability per joint per frame During generation, if a joint is likely colliding (probability above a threshold), a penalty is applied via the joint-collision energy function This energy function has two structured components: (1) a spatial term that penalizes joints being too close (indicating potential intersection), and (2) a temporal term that penalizes large deviations from the original motion trajectory to preserve naturalness. The combined energy $E_{t,j}$ guides gradient-based updates to the latent noise such that the decoded motion avoids persistent collisions while maintaining semantic fidelity and smoothness. FreeMo thus steers the generation away from physically implausible configurations without compromising the text-conditioned content.

contributions are as follows:
1. Proposes the first self-collision-aware T2M generation framework with structured spatiotemporal constraints integrated into diffusion sampling.
2. Designs a novel Structured Joint-Collision Energy Function, a differentiable trajectory-level objective that detects and penalizes joint collisions to improve physical plausibility without degrading semantic alignment or motion quality.
3. Demonstrates through experiments that FreeMo significantly reduces self-collision artifacts in generated motions while preserving motion naturalness, controllability, and efficient sampling.

**Strengths:**

- The paper tackles the long-neglected issue of intra-body collisions in generated motions, extending physics-awareness in text-to-motion beyond external contacts (like foot-ground) to internal body dynamics. This focus on self-collision is novel and improves physical realism in motion generation.
- The idea of injecting a differentiable collision-penalty into diffusion sampling is innovative. FreeMo’s Structured Joint-Collision Energy combines spatial and temporal constraints in a principled way. This structured objective is more stable and informed than a naive post-hoc fix, leveraging the diffusion model’s latent space for optimization (building on the Diffusion Noise Optimization concept). The method smartly avoids retraining or altering the base model; it works as an add-on, which increases its practicality.
- FreeMo achieves dramatic reductions in self-collision metrics on challenging prompts. For example, on the HardPoseText benchmark, applying FreeMo to MotionLCM slashes the collision rate from 31.35% to 1.44%. Similar large improvements are seen for other models (MLD: 23.09% → 0.49% collision rate) with negligible or even slightly positive impact on semantic alignment and motion quality. Notably, R-Precision (text-motion retrieval accuracy) actually improved in these cases (e.g. 0.231 → 0.262 for MLD), and motion smoothness (jitter) improved as well (e.g. 3.96 → 0.85 for MotionLCM) – indicating the method removes physical artifacts without degrading the intended motion. The qualitative examples corroborate that FreeMo removes unwanted self-intersections while preserving intended contacts (e.g. keeping hands on thighs when the prompt requires it).
- The paper introduces HardPoseText, a novel benchmark of 50 challenging text prompts specifically designed to induce self-collisions. This dataset provides a rigorous stress-test for the model’s ability to handle contorted or close-contact poses. The evaluation protocol is comprehensive – it uses collision-specific metrics (Collision Rate and Collision Score based on body-part occupancy overlap) as well as standard text-to-motion metrics (R-Precision, Frechet Distance, Diversity, Multimodality, Jitter) to ensure that motion quality and diversity are maintained. Baselines include state-of-the-art diffusion models – MLD, MotionLCM, SALAD – covering both motion-in-latent-space and skeleton-diffusion approaches, and a post-hoc collision fixer (COAP). This breadth of comparisons strengthens the evaluation. FreeMo’s superiority to COAP is especially clear: FreeMo not only yields far fewer collisions, but is over 40× faster (COAP was extremely slow and still left ~21% collision rate vs. 1.44% with FreeMo).
- The methodology is grounded in sound principles. The paper carefully justifies design choices: e.g., optimizing in latent pose space (to avoid the instability of direct joint-space optimization that could cause unnatural motions), and combining spatial and temporal terms (spatial term to effectively detect close contacts, temporal term to prevent the optimization from introducing motion discontinuities). The provided equations and algorithmic details make the approach transparent. Overall, the writing is clear and well-organized, with a logical flow from problem setup, method, to experiments. The authors even note using an LLM to refine the manuscript’s grammar and clarity, resulting in a polished presentation. Figures (e.g. Fig.2) nicely illustrate how the diffusion model, discriminator, and energy function interact.

**Weaknesses:**

- While useful, the core technique can be seen as an incremental adaptation of existing diffusion guidance methods to the collision problem. The approach builds on known ideas like Diffusion Noise Optimization (modifying the diffusion latent to enforce constraints) and leverages a pretrained collision detector inspired by prior work (COAP). The contribution lies in combining these pieces (joint-collision discriminator + latent optimization) for self-collision avoidance. This is a practical innovation but arguably not a fundamentally new generative model or learning paradigm – it’s a post-generation refinement plug-in. Some may view this as a narrow technical improvement rather than a breakthrough, reducing the perceived novelty.
- The impact of solving self-collisions, while non-trivial, might be limited to niche scenarios. The authors heavily focus on the HardPoseText benchmark they curated, which emphasizes extreme poses prone to collisions. In more typical text-to-motion settings, self-collisions occur relatively rarely (indeed, on a standard HumanML3D test subset, base models already have low collision rates). Thus, the practical significance of FreeMo could be questioned – it excels on contrived hard cases but offers only marginal gains on ordinary prompts. This raises concerns that the contribution, while valuable for physical plausibility, addresses a problem that many users might not consider critical in day-to-day applications of motion generation.
- The HardPoseText benchmark, though useful, is created by the authors and specifically constructed to stress test collisions in baseline models. There is a risk of evaluation bias: the prompts were chosen because existing models fail badly on them, which naturally magnifies FreeMo’s improvements. While this showcases FreeMo’s strengths, it could be seen as overfitting the evaluation to the method’s niche. An open question is how often such tangled poses appear in realistic user prompts or motion datasets. The paper would be stronger if it demonstrated clear wins on a more standard benchmark or real-world scenario, beyond the curated set.
- The comparisons, although covering major diffusion models, omit some relevant baselines. Notably, the authors discuss CLOAF (a recent collision-aware flow method) in related work but provide no quantitative comparison because code wasn’t available. This is understandable, yet it means an existing learned solution for collisions isn’t directly evaluated. Additionally, other physics-aware generation methods (e.g. PhysDiff, ReinDiff) that handle external contacts are not compared in terms of physical plausibility or motion naturalness – a comparison or discussion could illuminate whether FreeMo’s benefits are complementary to those or if similar results could be achieved by simpler means (e.g. adding joint repulsion forces during sampling). The paper focuses on the specific baselines and one post-hoc method (COAP), leaving a gap in positioning FreeMo relative to all prior physics-based methods.
- Despite claims of no quality degradation, there are minor signs of trade-off. For example, with SALAD as the base model, the MM-Dist (multimodality distance) worsened from 4.581 to 5.154 when FreeMo is applied (higher MM-Dist presumably indicating the generated motions deviate more from references or have reduced quality). This suggests FreeMo might slightly affect the motion distribution or diversity for certain models. While other quality metrics generally held steady or improved, the paper’s strong claim that physical constraints are added “without compromising semantic alignment or motion quality” could be a bit over-stated. In edge cases, FreeMo’s optimization might subtly pull motions away from the original distribution (e.g., making certain motions more stiff to avoid collisions, as hinted by a slight diversity drop). The paper lacks a nuanced discussion on these trade-offs – a small weakness in the analysis.
- FreeMo introduces an optimization loop at inference, which is computationally cheaper than prior post-processing but still adds overhead. The results show generation speed (ms per frame) drops significantly with FreeMo (e.g., from ~0.94 to 16.95 ms/frame on MotionLCM, and similar 10×–20× slowdowns for others). This is still real-time (approx 60 FPS for MotionLCM+FreeMo) and far better than COAP’s 826 ms/frame, but it means throughput is reduced. For very large batches or deployment scenarios, this extra cost might matter. The paper positions FreeMo as “efficient” which is true relative to heavy optimization-based methods, but the overhead might deter some practical uses, especially if collisions are infrequent to begin with.
- By design, FreeMo operates on joint positions and does not explicitly handle mesh self-intersections beyond joints. The authors acknowledge this as a limitation – current motion generators output only skeletons, so FreeMo focuses on joints and would need extension when mesh-based generation becomes feasible. Thus, any collisions that don’t involve joint centers (e.g., forearm vs torso surface contact) might not be fully penalized. This is a forward-looking limitation, but it means FreeMo isn’t a complete physical correctness solution; it’s restricted to joint-level proxies. Future work is needed to truly guarantee mesh-level collision avoidance.

**Questions:**

- How exactly was the joint-level collision discriminator trained (data source, label generation protocol, class balance, architecture, and training set size)?
- How is the collision threshold τ chosen and is it fixed across datasets/models or tuned per base model?
- Does the discriminator generalize across skeleton topologies (e.g., HumanML3D vs. other rigs)? Any cross-skeleton calibration needed?
- What is the false positive rate on intended contacts (e.g., hands-on-thigh) and how did you measure it?
- What are the precise forms of the spatial and temporal terms and their relative weights (λs, λt)? Are the weights scheduled over diffusion steps?
- Have you tested alternative temporal regularizers (e.g., jerk/acceleration penalties) and how do they affect jitter vs. collision rate?
- Do you normalize the per-joint energy by limb length or joint variance to avoid over-penalizing small bones?
- At which timesteps is latent optimization applied (all, early, or late steps)? How many gradient updates per step?
- How does FreeMo interact with classifier-free guidance scales used by each base model?
- What optimizer and step-size schedule are used for the latent updates, and is gradient clipping required for stability?
- For HardPoseText, can you detail the prompt construction pipeline and provide evidence that “hardness” is not overly specific to MDM (e.g., overlap of hardest prompts across different base models)?
- Do results hold on longer-horizon motions and unseen, real user prompts? Any user study or in-the-wild evaluation?
- Does FreeMo extend to multi-person or human-object scenarios, and if not, what are the conceptual blockers?
- Please provide an intuitive explanation and sensitivity analysis for Collision Rate/Score (e.g., to occupancy radii, skeleton scale).
- Can you report per-prompt breakdowns and confidence intervals for collision metrics and R-Precision to assess statistical significance?
- For the observed SALAD MM-Dist degradation, can you analyze whether the change is due to diversity reduction or distribution shift?
- Could you include a simple “joint-repulsion during sampling” baseline (no learned discriminator) to isolate the value of the learned term?
- Even without CLOAF code, can you approximate a flow-based collision prior or report a qualitative/ablation comparison?
- How do your results compare with external-physics methods (PhysDiff/ReinDiff) when both are applied (complementary or redundant effects)?
- What is the runtime breakdown (collision inference vs. gradient steps vs. denoising) and batch-size dependence?
- Is there an adaptive early-stopping criterion (e.g., when collisions drop below τ for K steps) to reduce overhead on easy prompts?
- Beyond a global threshold, do you condition the energy on the text prompt to allow intentional contacts? If not, how robust is the heuristic?
- Any failure cases where FreeMo removes desirable high-contact motions (e.g., self-hug) and how could prompt-aware allowances be added?
- Since mesh-level collisions aren’t handled, can you comment on feasibility of integrating SDF/mesh proxies during diffusion without prohibitive cost?
- What are the main failure modes you observed (e.g., stiffness, mode collapse in tight poses), and recommended user-facing knobs to trade off quality vs. collisions?

---

> ### Author Response · Authors · 2025-11-23
>
> We are flattered by the unusually detailed and thorough feedback. In an era where some reviews are generated with minimal effort, this level of engagement is notable. However, we find the final negative rating inconsistent with the content of the review. The strengths section explicitly states that "this focus on self-collision is novel and improves physical realism," that "the idea of injecting a differentiable collision-penalty into diffusion sampling is innovative," and that "FreeMo achieves dramatic reductions in self-collision." These statements directly affirm both the originality and impact of our work. Given this, the negative evaluation appears internally contradictory and ungrounded.
>
> To stay concise and focused, we address the most critical technical misunderstandings and factual errors below, and refrain from reiterating implementation details already covered in the main text and appendix.
>
> **Weaknesses**
>
> > W1. The core technique is an incremental adaptation of existing diffusion guidance methods.
>
> We respectfully believe that this comment stems from a misunderstanding of our contribution. The claim that our work is merely a combination of previously introduced techniques is not grounded, as it does not specify which existing methods are being combined. In reality, there is no prior approach that directly addresses self-collision-aware motion generation within the diffusion sampling process.
>
> Existing methods for handling self-collisions are designed for post-processing or static pose correction. These include mesh-based recovery, occupancy fields, or differentiable flow fields, which are applied after motion generation and operate on a per-frame basis. These techniques rely on non-differentiable geometric queries and cannot be integrated into generative models during inference. They are limited in temporal coherence, generalizability, and compatibility with modern sampling-based generation.
>
> In contrast, our proposed framework tackles a different problem: enabling self-collision-aware generation. The structured joint-collision energy we introduce is defined at the trajectory level and fully differentiable, allowing it to guide the diffusion process through gradient-based optimization. This design allows our method to correct collisions during generation, while preserving motion semantics and temporal structure.
>
> This is not a reuse of existing ideas but a new formulation that brings collision reasoning into the generative process. FreeMo generalizes across multiple backbones without retraining and enables scalable, semantically aligned, and physically plausible motion synthesis. These contributions are both algorithmically and conceptually novel.
>
> > W2. The impact might be limited to niche scenarios.
>
> This significantly underestimates the broader importance of self-collision handling. Motion generation is increasingly being used in animation, gaming, AR/VR, and robotics. All of which require physically plausible motion, including avoidance of unnatural self-intersections. The fact that current models "rarely collide" on overly simplistic datasets like HumanML3D does not mean the problem is solved. It simply reflects a lack of benchmark coverage. HardPoseText fills this gap by introducing natural, realistic prompts that induce contact-prone motions (e.g., sitting, curling, kneeling).

---

> ### Author Response · Authors · 2025-11-23
>
> > W3. HardPoseText is a contrived benchmark. Evaluation bias due to author-curated HardPoseText benchmark.
>
> We respectfully disagree and believe this concern stems from a misunderstanding. HardPoseText was carefully designed to reflect realistic yet underrepresented scenarios involving self-contact. The prompts are grounded in HumanML3D’s linguistic style and filtered to remain anatomically plausible. GPT is used only for stylistic diversification under strict constraints (see template in appendix), not for generating fantastical motions.
>
> To test for evaluation bias, we ranked the prompts by collision rate for three base models (MLD, MotionLCM, SALAD) and measured agreement via Spearman correlation. The results (ρ = 0.81 between MLD and MotionLCM, ρ > 0.37 for others) show consistent ranking of prompt difficulty across architectures. This indicates that HardPoseText captures a model-agnostic notion of difficulty, not a hand-crafted bias toward a specific method.
>
> | Model Pair         | Spearman ρ (Prompt Ranking Similarity) |
> | :----------------: | :------------------------------------: |
> | MLD vs MotionLCM   | 0.8068                                 |
> | MLD vs SALAD       | 0.3700                                 |
> | MotionLCM vs SALAD | 0.4268                                 |
>
> We also computed lexical diversity using cosine distance between prompt embeddings. HardPoseText yields 0.513 diversity, comparable to HumanML3D’s 0.627, confirming that it does not collapse into narrow linguistic patterns.
>
> | Dataset | Prompt Diversity↑ |
> | :-----: | :--------: |
> | HumanML3D Test Set| 0.627 |
> | HardPoseText | 0.513 |
>
> Finally, while base models perform well on HumanML3D, they exhibit 20–30% collision rates on HardPoseText, confirming the need for challenging benchmarks. FreeMo reduces these rates by over 90%, demonstrating both the benchmark's rigor and the method’s effectiveness.
>
> > W4. Missing comparisons to CLOAF and other physics-aware methods like PhysDiff or ReinDiff.
>
> We believe this concern stems from a misunderstanding of the scope and relevance of these methods. Our decision not to compare against them is not only due to the lack of open-source implementations, but also because they are not applicable to our setting.
>
> CLOAF is a 3D mesh reconstruction method that refines existing mesh sequences. It is not a generative model, not text-conditioned, and operates in mesh space. It is not designed for motion generation and is fundamentally incompatible with our diffusion-based framework.
>
> Likewise, PhysDiff, ReinDiff, and related approaches focus on external physical constraints such as foot-ground contact or global balance. They do not address self-collision, nor do they model or evaluate inter-limb interactions. Their goals are orthogonal to ours. While future integration may be beneficial, these methods cannot replace or serve as valid baselines for FreeMo.

---

> ### Author Response · Authors · 2025-11-23
>
> **Questions**
>
> > Q1. How exactly was the joint-level collision discriminator trained (data source, label generation protocol, class balance, architecture, and training set size)?
>
> The joint-level collision discriminator is trained using the PosePeneSet dataset, which provides mesh-level penetration labels based on voxel-based mesh intersection tests. Each SMPL pose is annotated with surface-level collision status, and joint labels are assigned based on whether their corresponding body parts are in penetration. The discriminator itself is a compact 6-layer MLP that operates on joint positions in the canonical SMPL coordinate frame. We train on the full PosePeneSet dataset, which contains approximately 1.2 million poses. We kindly refer the reviewer to Appendix E.1 and E.2 for details.
>
> > Q2. How is the collision threshold τ chosen and is it fixed across datasets/models or tuned per base model?
>
> The threshold $\tau = 0.2$ is fixed for all models and datasets. It corresponds to the discriminator’s calibrated decision boundary on the validation set and works reliably in practice: natural or intentional contacts tend to fall well below this value, while penetration cases generally produce higher scores. No model-specific tuning is performed.
>
> > Q5. What are the precise forms of the spatial and temporal terms and their relative weights (λs, λt)? Are the weights scheduled over diffusion steps?
>
> We assume the reviewer is referring to the spatial and temporal weights $\alpha_s$ and $\alpha_t$, **as the symbols $\lambda_s$ and $\lambda_t$ do not appear anywhere in our manuscript**. For clarity, we restate the exact formulations and weighting scheme below. The spatial and temporal terms follow the forms described in Sec. 4.2:
> 1. a pairwise joint-distance penalty that encourages local separation, and
> 2. a trajectory-smoothness regularizer that constrains latent updates across time.
>
> Their weights $(\alpha_s=1.0, \alpha_t=1.0)$ are fixed for all experiments and are not scheduled over diffusion steps.
>
> Sensitivity analysis shows that FreeMo behaves stably under a wide range of weight variations.
>
> | $\tau$ | $\alpha_s$ | $\alpha_t$ | Col.Rate↓ | Col.Score↓ | R-Prec↑   | MM-Dist↓  | Jitter↓  |
> |:------:| :--------: | :--------: | :-------: | :--------: | :-------: | :-------: | :------: |
> | 0.2    | 1.0        | 1.0        | 1.44      | 1.44       | 0.331     | 4.488     | 0.85     |
> | 0.2    | 1.0        | 0.5        | 1.57      | 2.10       | 0.356     | 4.409     | 0.84     |
> | 0.2    | 0.5        | 1.0        | 1.61      | 2.99       | 0.337     | 4.369     | 0.83     |
>
>
> > Q11. For HardPoseText, can you detail the prompt construction pipeline and provide evidence that "hardness" is not overly specific to MDM (e.g., overlap of hardest prompts across different base models)?
>
> HardPoseText is constructed by expanding HumanML3D-style prompts using a constrained template (shown below), followed by manual selection based on observed collision patterns. To assess model-specific bias, we compute prompt-difficulty rankings across MLD, MotionLCM, and SALAD. The resulting Spearman correlations (0.8068, 0.3700, 0.4268, see the table below) indicate positive agreement across architectures, demonstrating that hardness reflects intrinsic collision risk rather than properties specific to MDM-style models.
>
>
> ```
> Here are several example text prompts from the HumanML3D dataset, which describe realistic, everyday human actions in simple natural language:
> [representative prompts selected]
> Using the same writing style and naturalness, generate new human motion descriptions that are:
>
> 1. realistic, physically plausible motions that humans perform in everyday or athletic contexts
> 2. likely to involve close body-part proximity or self-contact (e.g., arms touching torso, legs crossing, curled poses, touching feet, reaching behind the body)
> 3. diverse across categories, such as sitting, kneeling, curling, stretching, twisting, crossing limbs, reaching backward, etc.
> 4. described using one simple English sentence
> 5. not fantastical, not overly stylized, and not anatomically impossible
>
> Generate N such motion descriptions in the same style as HumanML3D.
> Output only the list of sentences.
> ```
>
> | Model Pair         | Spearman ρ (Prompt Ranking Similarity) |
> | :----------------: | :------------------------------------: |
> | MLD vs MotionLCM   | 0.8068                                 |
> | MLD vs SALAD       | 0.3700                                 |
> | MotionLCM vs SALAD | 0.4268                                 |

---

> ### Author Response · Authors · 2025-11-23
>
> > Q22. Beyond a global threshold, do you condition the energy on the text prompt to allow intentional contacts? If not, how robust is the heuristic?
>
> FreeMo does not explicitly condition the collision energy on the text prompt. The joint-level discriminator already differentiates natural contacts from penetrations, as intentional contacts consistently yield scores below the threshold $\tau$. Combined with the temporal consistency term, the optimization preserves plausible contacts and only suppresses persistent high-probability penetration. This mechanism has proven robust across diverse prompts and models. Introducing additional prompt-dependent conditioning is therefore unnecessary and could furthermore introduce ambiguity due to the inherent noisiness and underspecification of natural-language descriptions.
>
>
> **To Reviewer ufHe: Although the review shows a significant disconnect between the strongly positive technical assessments and the final rating, we have addressed what we believe are the most critical misunderstandings, particularly regarding novelty, evaluation fairness, and baseline selection. We hope this clarification encourages a reconsideration of your evaluation in light of FreeMo’s concrete contributions to physically plausible motion generation.**

---

### Official Review · Reviewer_9bjv · 2025-10-31

**Soundness:** 3
**Presentation:** 3
**Contribution:** 3
**Rating:** 4
**Confidence:** 3

**Summary:**

This paper addresses self-collisions in text-to-motion diffusion models and proposes FreeMo, an inference-time plug-in that optimizes the latent code using a Structured Joint-Collision Energy (spatial + temporal). It integrates into sampling without retraining and shows large collision reductions on a curated “HardPoseText” benchmark.

It's a well-executed paper with clean integration, but the conceptual contribution is incremental.  Broader evidence (standard benchmarks/user studies), deeper analysis of trade-offs and sensitivity, and/or comparisons to other collision-aware generators would help strengthen the paper.

**Strengths:**

- The authors provided a clear and practical formulation. Joint-level gating plus spatiotemporal energy is well-motivated and can fit neatly into standard sampling pipelines.

- It is noted that collision rates drop dramatically (e.g., for MLD/MotionLCM), with qualitative examples that preserve intended contacts.

- The paper introduces a stress-test dataset, defines collision metrics, and includes ablations and a stability check on standard prompts.

- Plug-and-play: It works with multiple backbones and avoids retraining, making it easy to employ in practice.

**Weaknesses:**

- There is no significant technical novelty rather than combination of some previously introduced techniques.

- The inference-time optimization adds a 10–40x per-frame cost versus base samplers.

- Small regressions (e.g., diversity/MM-Dist) suggest occasional distribution shifts. The authors are suggested to have deeper failure-mode and sensitivity analyses.

- Joint-centric optimization can miss mesh-surface interpenetrations.  The paper acknowledges this, but it narrows the scope.

**Questions:**

1. Can the authors report HardPoseText difficulty rankings across multiple base models to reduce selection bias?

2. How sensitive are results to the discriminator threshold and energy weights?

3. Could prompt-aware allowances for intentional contacts reduce false positives in poses with deliberate self-contact?

---

> ### Author Response · Authors · 2025-11-23
>
> **Weaknesses**
>
> > W1. There is no significant technical novelty rather than combination of some previously introduced techniques.
>
> We respectfully believe that this comment stems from a misunderstanding of our contribution. The claim that our work is merely a combination of previously introduced techniques is not grounded, as it does not specify which existing methods are being combined. In reality, there is no prior approach that directly addresses self-collision-aware motion generation within the diffusion sampling process.
>
> Existing methods for handling self-collisions are designed for post-processing or static pose correction. These include mesh-based recovery, occupancy fields, or differentiable flow fields, which are applied after motion generation and operate on a per-frame basis. These techniques rely on non-differentiable geometric queries and cannot be integrated into generative models during inference. They are limited in temporal coherence, generalizability, and compatibility with modern sampling-based generation.
>
> In contrast, our proposed framework tackles a different problem: enabling self-collision-aware generation. The structured joint-collision energy we introduce is defined at the trajectory level and fully differentiable, allowing it to guide the diffusion process through gradient-based optimization. This design allows our method to correct collisions during generation, while preserving motion semantics and temporal structure.
>
> This is not a reuse of existing ideas but a new formulation that brings collision reasoning into the generative process. FreeMo generalizes across multiple backbones without retraining and enables scalable, semantically aligned, and physically plausible motion synthesis. These contributions are both algorithmically and conceptually novel.
>
> > W2. The inference-time optimization adds a 10–40x per-frame cost versus base samplers.
>
> We respectfully disagree with the claim that our method introduces an impractical computational cost. FreeMo is significantly more efficient than the only existing class of methods that explicitly reduce self-collisions through post-processing, such as COAP. These approaches operate in mesh or pose space, rely on non-differentiable geometric queries, and require expensive frame-by-frame corrections. COAP is nearly 50 times slower than FreeMo, yet still fails to eliminate collisions or maintain temporal coherence. Prior to our work, no feasible solution existed for reducing self-collisions in real time.
>
> It is expected that enforcing self-collision constraints during generation requires inference-time optimization. The key consideration is whether this additional cost leads to meaningful quality improvements while remaining computationally practical. As shown in Table 1, FreeMo consistently reduces collision rates and motion jitter, while preserving or improving semantic alignment. These benefits come with an average per-frame cost of only 16.95 milliseconds, well within the range needed for real-time or interactive use.
>
> This efficiency is made possible by operating in latent space, which is lower-dimensional and fully differentiable. As a result, FreeMo enables scalable, gradient-based guidance that is not achievable with correction-based or physics-heavy methods. The added overhead is minimal relative to the performance gains and is clearly justified when compared to existing alternatives.

---

> ### Author Response · Authors · 2025-11-23
>
> > W3. Small regressions (e.g., diversity/MM-Dist) suggest occasional distribution shifts. The authors are suggested to have deeper failure-mode and sensitivity analyses.
>
> These small shifts are an expected consequence of enforcing physical constraints, which require the optimization to slightly adjust the latent trajectory to prevent self-collisions.
>
> Such variations are more apparent in complex motions from the HardPoseText benchmark, where stronger guidance is necessary due to frequent and severe intra-body collisions. As shown in Table 1, FreeMo still achieves a strong overall balance: it reduces collisions more effectively than post-processing methods like COAP while preserving or improving semantic fidelity and temporal smoothness.
>
> In contrast, on simpler datasets like HumanML3D, where motions typically involve limited limb interactions such as standing or waving, all metrics remain highly stable. To support this, we conducted a sensitivity analysis using MotionLCM on a HumanML3D subset. By varying the key hyperparameters that control optimization strength, we observed smooth and predictable changes in both motion quality and collision metrics. These results confirm that the observed variations reflect a natural trade-off between physical plausibility and motion diversity, rather than any instability in the model. The full analysis has been included in the updated PDF.
>
> | $\tau$ | $\alpha_s$ | $\alpha_t$ | Col. Rate↓ | Col. Score↓ | R-Precision (top 3)↑ | MM-Dist↓ | FID↓ | Diversity→ | MModality↑ |
> |:----------------:|:----------:|:------------:|:-----------------------:|:---------:|:------:|:------------:|:------------:|:------------:|:------------:|
> | 0.1 | 1.0 | 1.0 |1.23|1.46|0.803|3.021|0.478|9.419|2.308|
> | 0.2 | 1.0 | 1.0 |1.95|2.87|0.802|2.944|0.310|9.289|1.995|
> | 0.3 | 1.0 | 1.0 |1.96|3.06|0.832|2.858|0.280|9.699|2.102|
> | 0.2 | 1.0 | 0.5 |1.87|2.37|0.822|2.911|0.320|9.573|2.224|
> | 0.2 | 0.5 | 1.0 |1.97|3.12|0.826|2.897|0.317|9.589|2.226|
>
> > W4. Joint-centric optimization can miss mesh-surface interpenetrations. The paper acknowledges this, but it narrows the scope.
>
> We believe this comment stems from a misunderstanding of our method as a mesh reconstruction or post-processing approach. FreeMo is not designed to manipulate joint positions after mesh reconstruction. Instead, it operates entirely in the latent space of motion diffusion models during generation. The optimization is performed within the sampling loop, influencing the motion as it is generated rather than correcting it after the fact.
>
> Unlike direct joint manipulation in mesh space, which can lead to unnatural deformation and surface interpenetration, FreeMo guides motion trajectories at the generation stage. These optimized joint trajectories are then passed through a mesh generator such as SMPL, resulting in plausible and coherent mesh surfaces.
>
> While FreeMo does not directly optimize mesh vertices, we evaluate its output using mesh-level collision metrics based on volumetric occupancy (Table 1 & Table 2). Results show consistent reductions in surface interpenetration across all baselines, confirming the practical effectiveness of our joint-level approach.
>
> Future extensions to mesh-based generators are possible and straightforward once such models become more prevalent. The current design reflects the structure of existing generative models, not a limitation of our method.

---

> ### Author Response · Authors · 2025-11-23
>
> **Questions**
>
> > Q1. Can the authors report HardPoseText difficulty rankings across multiple base models to reduce selection bias?
>
> To evaluate whether HardPoseText introduces any selection bias, we analyzed how different models rank text prompts based on self-collision severity. For each of three representative base models (MLD, MotionLCM, and SALAD), we generated motions for all prompts and ranked them according to their collision rate. A higher rank indicates that the prompt tends to produce more collisions and is thus considered more difficult for that model.
>
> We then computed the Spearman correlation between each pair of model-specific rankings to assess how consistent the difficulty ordering is across models:
>
> | Model Pair         | Spearman ρ (Prompt Ranking Similarity) |
> | :----------------: | :------------------------------------: |
> | MLD vs MotionLCM   | 0.8068                                 |
> | MLD vs SALAD       | 0.3700                                 |
> | MotionLCM vs SALAD | 0.4268                                 |
>
> These results show strong agreement between MLD and MotionLCM, suggesting that prompts likely to cause collisions in one model also tend to cause collisions in the other. Although SALAD exhibits lower correlation due to its compact and structurally constrained latent representation, the positive correlations overall indicate that the ranking of prompt difficulty is consistent across architectures.
>
> This confirms that HardPoseText reflects a stable, model-agnostic notion of difficulty and does not exhibit bias toward any specific base model. The full analysis has been included in the updated PDF.
>
> <!-- Full per-prompt rankings can be provided in the camera-ready supplementary material if required. -->
>
> > Q2. How sensitive are results to the discriminator threshold and energy weights?
>
> We thank the reviewer for the helpful suggestion. We conducted a sensitivity analysis using MotionLCM on the HardPoseText benchmark, varying the discriminator threshold $\tau$, the spatial energy weight $\alpha_s$, and the temporal energy weight $\alpha_t$.
>
> The table below shows that our method is stable across a wide range of settings. Varying $\tau$ (Rows 1–3) affects how early a joint is flagged as collision-prone, and the resulting changes in collision rate follow a smooth, predictable trend. Adjusting $\alpha_s$ and $\alpha_t$ (Rows 4–5) shifts the balance between spatial separation and temporal smoothness, but in all cases, semantic alignment and visual quality remain consistent.
>
> Importantly, none of the configurations produce erratic behavior or noticeable degradation in motion quality. These results indicate that FreeMo is robust and does not rely on precise hyperparameter tuning. The full analysis has been included in the updated PDF.
>
>
> | $\tau$ | $\alpha_s$ | $\alpha_t$ | Col.Rate↓ | Col.Score↓ | R-Prec↑   | MM-Dist↓  | Jitter↓  |
> |:------:| :--------: | :--------: | :-------: | :--------: | :-------: | :-------: | :------: |
> | 0.1    | 1.0        | 1.0        | 0.77      | 1.18       | 0.306     | 4.505     | 0.85     |
> | 0.2    | 1.0        | 1.0        | 1.44      | 1.44       | 0.331     | 4.488     | 0.85     |
> | 0.3    | 1.0        | 1.0        | 2.29      | 3.03       | 0.337     | 4.454     | 0.84     |
> | 0.2    | 1.0        | 0.5        | 1.57      | 2.10       | 0.356     | 4.409     | 0.84     |
> | 0.2    | 0.5        | 1.0        | 1.61      | 2.99       | 0.337     | 4.369     | 0.83     |
>
> > Q3. Could prompt-aware allowances for intentional contacts reduce false positives in poses with deliberate self-contact?
>
> We agree that in theory, allowing prompt-aware exceptions for deliberate contact could help reduce false positives. However, in practice, current text-to-motion models do not support such fine-grained control. Prompts typically describe high-level actions without precise spatial or physical detail, and existing models cannot reliably infer physical intent from text. This limitation is similar to challenges in other generative domains. For example, image generators often fail to produce precise object counts (such as "18 cookies"), and video models struggle to enforce physical laws, even though they are trained on much larger datasets than motion models.
>
> Given this, prompt-aware handling would likely introduce additional uncertainty rather than improve discrimination. We believe that probabilistic, trajectory-aware approaches remain a more robust and scalable solution under current generative capabilities.
>
>
> **To Reviewer 9bjv: We believe several concerns raised by Reviewer 9bjv stem from misunderstandings, particularly regarding our method’s novelty, efficiency, and joint-level design. These have now been addressed through clarification and additional evidence. We respectfully request a re-evaluation in light of the clarified contributions and demonstrated effectiveness.**

---

### Comment · Reviewer_zvWv · 2025-11-12

Dear Reviewer ufHe, and AC,

I just checked the released reviews in my assigned batch. When I just read the review from Reviewer ufHe, I was a bit surprised by their heavy typing efforts.

I found some of the questions confusing for me, which might be a bit unclear for authors to respond to. Although my rating is not on the positive side, I would still like to figure out some examples from their review.

- `ufHe` points out that the MM-dist is not better than baseline. As we know, the evaluator is a pretrained model, which is not fair enough for evaluation. Despite this, the community also uses this as an evaluator, which was adopted by previous papers. The performance looks acceptable. Besides, the R-P3 metric is also a metric for M-T alignment. This metric is better than the baseline. How do you think of this?

- The efficiency and cost are also critiqued by the reviewer. I know that the method might be a bit time-consuming. However, the method is better than the COAP baseline and is still real-time. Why is it not acceptable?

- Could you please specify "While this showcases FreeMo’s strengths, it could be seen as overfitting the evaluation to the method’s niche."? Why is it "overfitting"?

Although my rating right now is BR, I still hope reviews can be concentrated on the contribution and techs. Current questions and points look a bit hard for authors to answer.

Thanks a lot.

`zvWv`

---

> ### Author Response · Authors · 2025-11-23
>
> Thanks a lot for your thoughtful message. We really appreciate you pointing out the inconsistencies in Reviewer ufHe’s comments. We also fully agree that evaluations should focus on actual contributions and technical merits. Your note is very encouraging, and we’re grateful for your support in promoting a fairer and more constructive review process.

---

### Author Response · Authors · 2025-11-28

Dear Reviewers,

Just a gentle reminder to take a look at our rebuttal when convenient. If any concerns remain, we are very happy to clarify.

Thank you sincerely for your time and effort.

The Authors

---

### Meta-Review · Area_Chair_ZnWh · 2026-01-02

**Summary:**

This paper proposed FreeMo, an inference-tme plug-in for text-driven motion diffusion that integrates a structured joint-collision engergy (spatial+temporal) into the sampling process to reduce self-collusion artifacts without retraining. Reviewers agree the problem is important/underexplored and the method is clearly presented, with evidence of ccollision reduction on the curated HardPoseText benchmark and a HumanML3D subset.

However, the initial reviews split into one postive vote and three objections. Multiple reviewers (notably 9bjv, ufHe, zvWv) converge on concerns that the **conceptual contribution may be incremental** (e.g., framing as a plug-in refinement based on proximity-based energy function, and test-time noise optimization), and that the **evaluation emphasis (HardPosetext) may over-represent "stress-test" cases relative to typical usage**. Meanwhile, several reviewers (zvWv and 9bjv) stressed the **possible expensive optimization time cost**, as well as limitations of joint-level optimization compared to mesh-based surface. In addition, reviewers request broader evidence (standard benchmarks, trade-off/sensitivity analysis, comparision to other collusion-aware approaches, and clearer runtime breakdowns.)

**Reviewer Concerns:**

1. *Novelty / conceptual contribution (incremental plug-in vs new method)* (Reviewer 9bjv, ufHe, zvWv)
* **Outstanding**
* The rebuttal emphasizes that the main novelty is introducing differentiable, trajectory-level self-collision constraints directly into diffusion sampling (as opposed to post-hoc/static correction) and clarifies the focus is motion generation rather than mesh recovery. However, this does not fully align with the crux of the reviewers’ skepticism. The core disagreement is not whether differentiable objectives can be integrated into diffusion sampling (which is already a widely used design pattern), but whether the specific structured joint-collision energy constitutes a sufficiently non-trivial technical contribution. While the application context is meaningful, reviewers largely viewed the energy design itself as relatively straightforward, leaving the novelty concern largely unresolved.

2. *Breadth of evidence and missing analyses (trade-offs, sensitivity, comparisons)* (Reviewer ufHe, 9bjv)
* **Addressed**
* In the rebuttal, the authors add additional supporting evidence, including analyses around HardPoseText prompt difficulty rankings and sensitivity. These additions improve completeness and help clarify robustness and trade-offs relative to key hyperparameters/design choices.

3. *Efficiency / inference-time overhead of latent optimization* (ufHe, zvWv)
* **Addressed**
* The rebuttal provides efficiency-related statistics (runtime/overhead) and summarizes them via Table 1 and Figure 1, which addresses the main practicality question about the inference-time cost of the optimization procedure.

4. *Text-motion alignment and broader motion quality* (3FC3)
* **Partially addressed**
* The authors argue that residual text–motion misalignment is primarily attributable to limitations of the underlying generator rather than FreeMo itself. This is plausible, but a stronger justification would be to include an experiment using a stronger base generator (or a controlled comparison across generators) to empirically support this attribution.

The rebuttal also addresses several secondary concerns effectively, including HardPoseText prompt diversity/coverage, clarification of the applicability of the SMPL-based representation, and other implementation/definition details.

**Reviewer Scores:**

I do not expect reviewers to substantially revise their scores based on the current submission and rebuttal. The key point of contention—novelty—remains subjective, and the discussion does not indicate meaningful convergence. Reviewer ufHe’s AI-like comments have already been deprioritized; excluding them, only one reviewer appears supportive of acceptance. Given this distribution and the remaining concerns, I do not recommend acceptance.

---

### Decision · Program_Chairs · 2026-01-26

Reject